# Characterization of *Anopheles gambiae* immune cells through genetic and functional immunophenotyping

George-Rafael Samantsidis , Hyeogsun Kwon  & Ryan C. Smith  ✉

Mosquito immune cells, or hemocytes, are integral components of the innate immune responses that define vector competence. To date, the characterization and functional classification of hemocytes has been hindered by the limited availability of genetic resources. Here, we map the composition of mosquito hemocytes by engineering five transgenic *Anopheles gambiae* lines that express fluorescent proteins under the control of candidate hemocyte promoters. We characterize these five transgenic lines through gene expression and microscopy-based approaches, and examine mosquito immune cell populations by leveraging advanced spectral imaging flow cytometry. We classify mosquito hemocytes into twelve distinct populations based on size, granularity, and ploidy, while defining these hemocyte subtypes based on their phagocytic capacity and the expression of genetic markers. By simultaneously analyzing these morphological and genetic properties, our work highlights the complexity and plasticity of mosquito hemocytes and provides the foundation for deeper investigations into their roles in immunity and pathogen transmission.

Immune cells are crucial components of the immune system in all Metazoa[1], playing key roles in limiting infection, pathogen clearance, developmental regulation, and maintaining tissue homeostasis[2,3]. While vertebrate immune cells contribute to both innate and adaptive immune responses, invertebrates solely rely on innate immune mechanisms, where immune cells are essential to combat pathogen infections, maintain homeostasis, and ensure host survival[1,4]. Much of our understanding of insect cellular immunity has relied on studies in *Drosophila*[5,6], where the genetically tractable system and extensive genetic resources have provided an important foundation for our understanding of cellular immune function and hematopoiesis in other insect systems.

In mosquitoes, traditional classification based on morphological properties has led to the identification of three major immune cell (hemocyte) subtypes, namely granulocytes, oenocytoids, and prohemocytes[7–9]. Granulocytes, analogous to mammalian macrophages, are phagocytic cells that primarily act as immune sentinels regulating immune homeostasis and pathogen elimination.

Oenocytoids are most often recognized for their role in the production of prophenoloxidases (PPO)[10]. Finally, prohemocytes are presumed precursor cells thought to differentiate into the granulocyte and oenocytoid lineages[7,11,12]. While previous studies using lectin-conjugated stains[7,13–15] or lipophilic dyes[16–18] have enhanced the visualization of these mosquito immune cell populations, these methods provide general hemocyte staining and do not adequately resolve mosquito hemocyte subtypes[7,16]. In addition, while the application of clodronate liposomes in mosquitoes has enabled techniques to examine the functional contributions of phagocytic granulocyte populations[11,19–21], we still lack tools to evaluate the function of non-phagocytic cell types. As a result, our understanding of mosquito immune cell populations has primarily been limited to morphological observations, leading to discrepancies in cell classifications[22] and hemocyte numbers[8].

Immunophenotyping via flow cytometry has proven invaluable for studying the dynamics and heterogeneity of vertebrate immune cell populations[23,24]. In mosquitoes, flow cytometry has been used to

Department of Plant Pathology, Entomology and Microbiology, Iowa State University, Ames, Iowa, USA. ✉e-mail: smithr@iastate.edu

identify phagocytic cell populations[19,25] or to identify changes in cell morphology and ploidy in response to blood feeding[14,26]. However, further applications of flow cytometry in mosquitoes have been constrained by the lack of specific cell markers or antibodies needed to define mosquito immune cell subpopulations. Recently, proteomic and single-cell studies have begun to address this limitation by identifying candidate hemocyte markers[11,27–29], while expanding our understanding of mosquito immune cell populations[11,29].

The development of genetic resources in *Drosophila* has significantly advanced our understanding of insect immune cells and hematopoiesis, enabling precise labeling, manipulation of gene expression, and genetic ablation[30–34]. However, the development of similar tools in *Anopheles* has been limited thus far. Attempts to utilize the *Drosophila* hemocyte-specific hemolectin promoter in *Anopheles gambiae* have been met with mixed success, resulting in the labeling of a subset of hemocytes only after blood feeding[35]. A previous study has also demonstrated the utility of the *Anopheles* PPO6 promoter to label mosquito hemocyte populations[36]. While these PPO6-labeled cells were profiled in a preliminary single-cell study[28], their phenotypic characteristics, abundance, and expression patterns in hemocyte subtypes has not yet been comprehensively reported. Despite these limitations of previous studies, the integral role of hemocytes in modulating vector competence to virus[37,38] and malaria parasite infection[13,15,19,39–41] strongly supports the need to develop genetic tools to further enhance our understanding of mosquito hemocyte function.

Here, we examine an extended list of presumed pan-hemocyte and granulocyte-specific promoters in *An. gambiae* to facilitate the visualization and characterization of mosquito immune cells. After generating transgenic lines for five hemocyte promoter constructs, we characterize their pattern of expression through qRT-PCR and microscopy. Using these genetic resources, we employ imaging spectral flow cytometry to define the mosquito immune cell landscape at high resolution. Through this approach, we identify twelve hemocyte subtypes based on physical and morphological properties, that are defined by the expression of genetic markers and their phagocytic properties, further resolving these mosquito immune cell subtypes. In summary, our study provides a strong foundation for genetic immunophenotyping in mosquitoes, advancing our understanding of mosquito immune cell biology and creating a foundation for future studies to examine the specific roles of hemocytes in mosquito-borne disease transmission.

## Results

### Developing transgenic mosquito lines to evaluate candidate hemocyte promoters

With the aim of establishing transgenic *An. gambiae* that specifically express fluorescent markers in hemocytes, we used previously published single-cell[11,28,29] and proteomic[27] datasets, as well as information from previous functional studies[7,19,36,42,43], to identify potential genes enriched across all hemocyte subtypes or specifically in granulocytes. As a result, we selected the promoter regions of NimB2 (AGAP029054), PPO6 (AGAP004977), and SPARC (AGAP000305) as putative "panhemocyte" promoters (Supplementary Fig. 1) and the promoters of LRIM15 (AGAP007045) and SCRASP1 (also known as Sp22D; AGAP005625) as putative granulocyte-specific promoters (Supplementary Fig. 1). For each gene, genomic fragments including the 5′ UTR and ~2000 bp upstream of the putative transcription start site were used to capture the putative regulatory regions of each candidate gene promoter (Supplementary Table 1). Five different piggyBac transposon constructs were generated containing the putative panhemocyte or granulocyte-specific promoters fused with CFP or GFP, respectively (Supplementary Fig. 1). Each construct was successfully integrated into the *An. gambiae* genome, as confirmed by splinkerette PCR, with at least two transgenic lines generated for each promoter construct (Supplementary Fig. 2).

To determine if the random integration of piggyBac caused position effects on promoter activity, we examined fluorescent marker expression across the different transgenic lines for each construct. We found significant differences in the three different lines generated with the PPO6-CFP construct, where the AP line displayed the highest expression. In contrast, the F1 line exhibited minimal CFP expression, and therefore, was not further evaluated (Supplementary Fig. 3). Similarly, expression differences were observed for the SCRASP1-GFP construct, with the LGAP line displaying significantly higher transgene expression than the F2B line (Supplementary Fig. 3). No significant differences in expression were observed between the individual lines for the SPARC-CFP, NimB2-CFP, or LRIM15-GFP constructs (Supplementary Fig. 3). Of note, while *NimB2* was found to be significantly enriched in previous transcriptomic and proteomic datasets[11,27,28], NimB2-CFP transgene expression was ~10-30 times lower than that of the PPO6-CFP or SPARC-CFP constructs suggesting that the regulatory regions used for the NimB2 construct may not be adequate to drive fluorescent marker expression (Supplementary Fig. 3).

### Molecular characterization of putative panhemocyte markers
Similar to previous studies with the PPO6 promoter[36], we observed PPO6-driven CFP fluorescence in circulating hemocytes of whole mount mosquito larvae and pupae (Fig. 1a). While we observe similar patterns of SPARC-driven CFP expression to that of PPO6 in larval and pupal hemocytes, the SPARC promoter also displayed visible fluorescence in the fat body of both developmental stages (Fig. 1b). Additional qPCR analysis demonstrates that there are comparable levels of expression of the CFP marker in larvae and adults for both the PPO6 (Fig. 1c) and SPARC (Fig. 1d) promoters, with PPO6 promoter driving slightly higher levels of *CFP* expression in adults (Fig. 1c).

To further validate these findings, we perfused individual mosquitoes from each transgenic line to examine the fluorescence activity in adult hemocytes. Microscopic observations of hemocytes from PPO6-CFP adult transgenic mosquitoes revealed the existence of two distinct immune cell populations (Fig. 1e), referred to as PPO6^low and PPO6^high, as previously described[19,28,36]. While most of the PPO6+ cells were classified as granulocytes by their phagocytic capacity, the remaining PPO6+ cells could not readily be classified as other hemocyte subtypes based on morphology (Fig. 1e). Similar observations were made for hemocytes perfused from SPARC-CFP mosquitoes, with varying CFP expression patterns detected among individual immune cells (Fig. 1f). Although the prevalence of SPARC+ cells consisted primarily of phagocytic granulocytes with unique elongated projections extending outward from the cellular body, a limited number of cells displayed different morphological features, that were smaller in size and circular shaped with concentric nuclei potentially representative of other hemocyte subtypes such as oenocytoids or prohemocytes (Fig. 1f). Unfortunately, NimB2-CFP mosquitoes failed to display CFP fluorescence in perfused hemocytes (Supplementary Fig. 4), consistent with the low expression levels of CFP in two different transgenic lines (Supplementary Fig. 3). Additional experiments to examine whether NimB2-CFP expression could be influenced by blood-feeding, similar to previously described hemocyte promoters[35], confirm the minimal levels of CFP expression under both naïve and blood-fed conditions (Supplementary Fig. 4). Together, these results confirm that the regulatory regions of the NimB2 promoter construct are inadequate to drive heterologous expression, and therefore, was excluded from further analysis.

To examine the hemocyte-specificity of each promoter, we examined CFP expression in perfused hemolymph and carcass tissues for the PPO6-CFP and SPARC-CFP transgenic lines. Similar to the endogenous expression of the hemocyte-specific gene NimB2[11,19,28], *CFP* expression was significantly enriched in hemocytes compared to carcass tissues for both the PPO6-CFP (Fig. 1g) and SPARC-CFP (Fig. 1h)

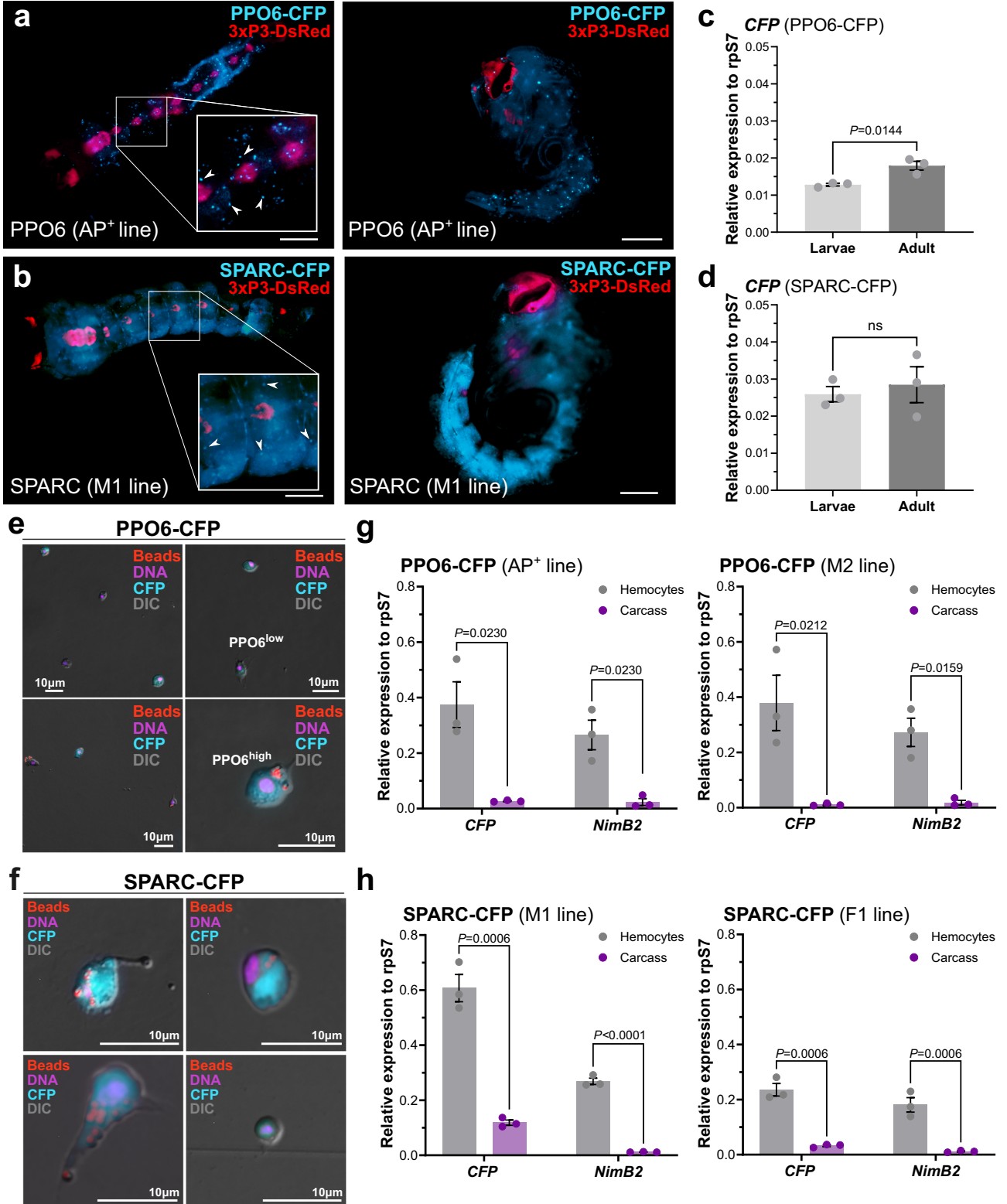

transgenic constructs, with similar patterns of expression among different transgenic lines. However, the enrichment of *CFP* expression in hemocytes was less pronounced in SPARC-CFP mosquitoes when compared to PPO6-CFP transgenics (Supplementary Table 2), suggesting that CFP may be expressed in other mosquito tissues such as the fat body (Fig. 1b). Additional experiments using clodronate-liposomes to deplete phagocytic granulocyte populations[11,19–21] provide further support for the enrichment of PPO6-CFP (M2 line) in

granulocyte populations (Supplementary Fig. 5). However, neither SPARC-CFP line displayed reduced CFP expression following clodronate liposome treatment (Supplementary Fig. 5), suggesting that the SPARC promoter drives expression in non-phagocytic hemocytes (resistant to clodronate treatment) or in other tissues beyond that of hemocytes. The latter, is supported by observations of CFP fluorescence in the fat body of larval and pupal stages in SPARC-CFP mosquitoes (Fig. 1b).

**Fig. 1 | Molecular characterization of the candidate panhemocyte markers, PPO6 and SPARC.** CFP fluorescence was examined in whole mount fourth-instar larvae and pupae from PPO6 (**a**) or SPARC (**b**) transgenic lines. Arrowheads highlight presumed hemocytes. Scale bars: 1 mm. Potential differences in *CFP* expression between larvae and adult mosquitoes were examined by qPCR for both PPO6 (**c**) or SPARC (**d**) transgenic lines. For (**c** and **d**), expression data are displayed as the mean ± SE from three independent replicates ($N = 3$, dots) of pooled ($n = $ -10) larvae or adult female mosquitoes, with values relative to rpS7 expression. Significance was determined using a two-tailed unpaired Student's *t*-test. Exact *P* values are displayed in the figure where significant. ns not significant. Additional ex vivo analysis of perfused hemocytes from PPO6 (**e**) or SPARC (**f**) transgenic lines using microscopy was used to examine hemocyte fluorescence. The injection of fluorescent beads (red) prior to perfusion indicates that both PPO6-CFP⁺ and SPARC-

CFP⁺ immune cell populations are comprised of phagocytic and non-phagocytic cells. Scale bars: 10 μm. Subpanels demonstrate different representative cell phenotypes. The specificity of *CFP* expression was examined in hemocytes and carcass samples via qPCR for the PPO6 (**g**) and SPARC (**h**) constructs across individual transgenic lines. The hemocyte-specific expression of *NimB2* was used as a positive control for gene expression analysis. Expression data are displayed as the mean ± SE from three independent replicates ($N = 3$, dots) of pooled hemolymph perfusions ($n = $ >30 adult female mosquitoes) or adult female mosquitoes ($n = $ -10) following perfusion (carcass), with values relative to rpS7 expression. Significance was determined using multiple two-tailed unpaired *t*-tests and a Holm–Šídák correction. Adjusted *P* values are displayed in the figure where significant. Source data are provided as a Source Data file.

In addition, using a commercially available CFP antibody and a previously described PPO6 antibody[19,26,43], we compared CFP expression with that of the endogenous PPO6 protein using immunofluorescence on perfused hemocytes from the PPO6-CFP line. Similar to previous observations[19,26], the majority (-80%) of fixed hemocytes were PPO6⁺ (Supplementary Fig. 6). Of these PPO6⁺ hemocytes, -58% of the total cells (or -73% of PPO6⁺ cells) were positive for both CFP and PPO6 staining (Supplementary Fig. 6). This suggests that CFP expression driven by the PPO6 promoter accurately represents the patterns of endogenous PPO6 expression. However, given that not all PPO6⁺ cells are CFP⁺, the provided regulatory regions of the PPO6 promoter used in our experiments may be suboptimal in driving detectable levels of CFP expression across all PPO6⁺ hemocyte populations. Alternatively, this discrepancy may result from the intercellular trafficking of PPO6 via extracellular vesicles between hemocyte populations as previously proposed[28].

## Molecular characterization of putative granulocyte markers

Granulocytes are central components of the mosquito innate immune responses that contribute to pathogen recognition and killing[19,40]. With previous studies featuring LRIM15 SCRASP1 prominently as granulocyte markers based on gene and protein expression analyses[11,27], we opted to examine these gene promoters for their ability to drive granulocyte-specific expression. In contrast to the PPO6 and SPARC transgenic lines, GFP fluorescence was not visibly detected in transgenic larvae of the LRIM15-GFP and SCRASP1-GFP lines (Fig. 2a, b). Additional experiments using qPCR to compare the levels of *GFP* between larvae and adults for each transgenic line verified these observations, highlighting the specificity of these promoters only to the adult stages (Fig. 2c, d). Perfusion of LRIM15-GFP transgenic mosquitoes followed by immunostaining confirmed GFP expression in phagocytic granulocyte populations, consisting of cells with high and low patterns of GFP fluorescence that were observed in both native and fixed conditions (Fig. 2e and Supplementary Fig. 7). In contrast with the LRIM15-GFP construct, SCRASP1-GFP hemocyte populations had much lower proportions of GFP-positive cells, with sizes ranging from approximately 3 to 10 microns (Fig. 2f and Supplementary Fig. 7).

The specificity of granulocyte-specific promoters was further evaluated by comparing *GFP* expression in perfused hemolymph with carcass tissues. While LRIM15 promoter activity was -3 times higher in hemocytes than in carcass tissues, there was no difference in GFP expression between tissues for the SCRASP1-GFP construct (Fig. 2g–i). To further confirm these observations, we again employed the use of clodronate liposomes to determine the effects of granulocyte depletion on *GFP* for each granulocyte promoter construct. While injections with clodronate-liposomes decreased *NimB2* expression by -60% in each strain, suggestive of phagocytic granulocyte depletion, clodronate treatment significantly reduced *GFP* expression in the LRIM15-GFP lines but not in the SCRASP1-GFP line (Supplementary Fig. 5). We view this limited effect of clodronate treatment on *GFP* expression in the SCRASP1-GFP line as the result of low activity levels of the

transgene, combined with the leaky expression in other tissues as indicated by qPCR (Fig. 2i). While we cannot exclude that there are low levels of expression in non-target tissues under the LRIM15 promoter, our results support that LRIM15 serves as a valuable granulocyte-specific marker.

## Hemocyte fluorescent markers reveal dynamic shifts in response to blood feeding

Previous studies have suggested that mosquito hemocytes are dynamic and undergo significant changes in response to blood-feeding[14,26,27,44]. For this reason, we examined the influence of blood-feeding on marker expression for each of our PPO6, SPARC, and LRIM15 hemocyte promoter constructs using microscopy and gene expression methods. Based on the low abundance of GFP⁺ cells and weak patterns of GFP expression (Fig. 2 and Supplementary Figs. 5 and 8), the SCRASP1 construct was not included in our further analysis.

The abundance of PPO6-CFP⁺ hemocytes remained stable between sugar-fed and 24 h post-blood feeding, representing -10% of total immune cells (Fig. 3a and Supplementary Table 3) and consistent with patterns of PPO6-driven *CFP* gene expression analysis (Supplementary Fig. 8). However, at 48 h post-feeding, PPO6⁺ cells displayed a small but significant increase in abundance (Fig. 3a), which can be attributed to an expansion in the proportions of PPO6^low populations (Supplementary Fig. 9). In contrast to the patterns observed in PPO6 immune cell populations, SPARC-CFP⁺ cells were more prevalent and displayed temporal oscillations in their abundance. We observed a significant increase in the proportions of CFP⁺ hemocytes from naive sugar-fed to 24 h post-blood meal (-45% to -63%), yet by 48 h post-feeding, SPARC⁺ cell proportions significantly declined to -33% (Fig. 3b and Supplementary Table 3). Despite this variation in cell populations, no changes in SPARC-driven *CFP* expression were measured (Supplementary Fig. 8). Of note, LRIM15-GFP⁺ cells displayed an inverse phenotype in response to blood-feeding, with a significant reduction of GFP⁺ cell proportions at 24 h post-blood feeding, before reverting back to baseline levels (-30% of cells) at 48 h post-feeding (Fig. 3c and Supplementary Table 3). This is further supported by a corresponding decrease in GFP expression in the LRIM15-AP line at 24 h post-blood feeding (Supplementary Fig. 8). Together, these data suggest that *An. gambiae* hemocyte populations are heterogeneous in nature and plasticity as they respond to physiological signals such as blood-feeding.

## Mosquito immune cells comprise multiple subtypes based on ploidy and morphology

While conventional flow cytometry has been previously used to demonstrate differences in the DNA content of mosquito immune cells[11,14,26], these studies have been limited by the lack of well-defined genetic markers and an inability to visualize cell heterogeneity in high resolution. With the advent of new technologies that combine spectral and imaging flow cytometry (IFC), and the development of the

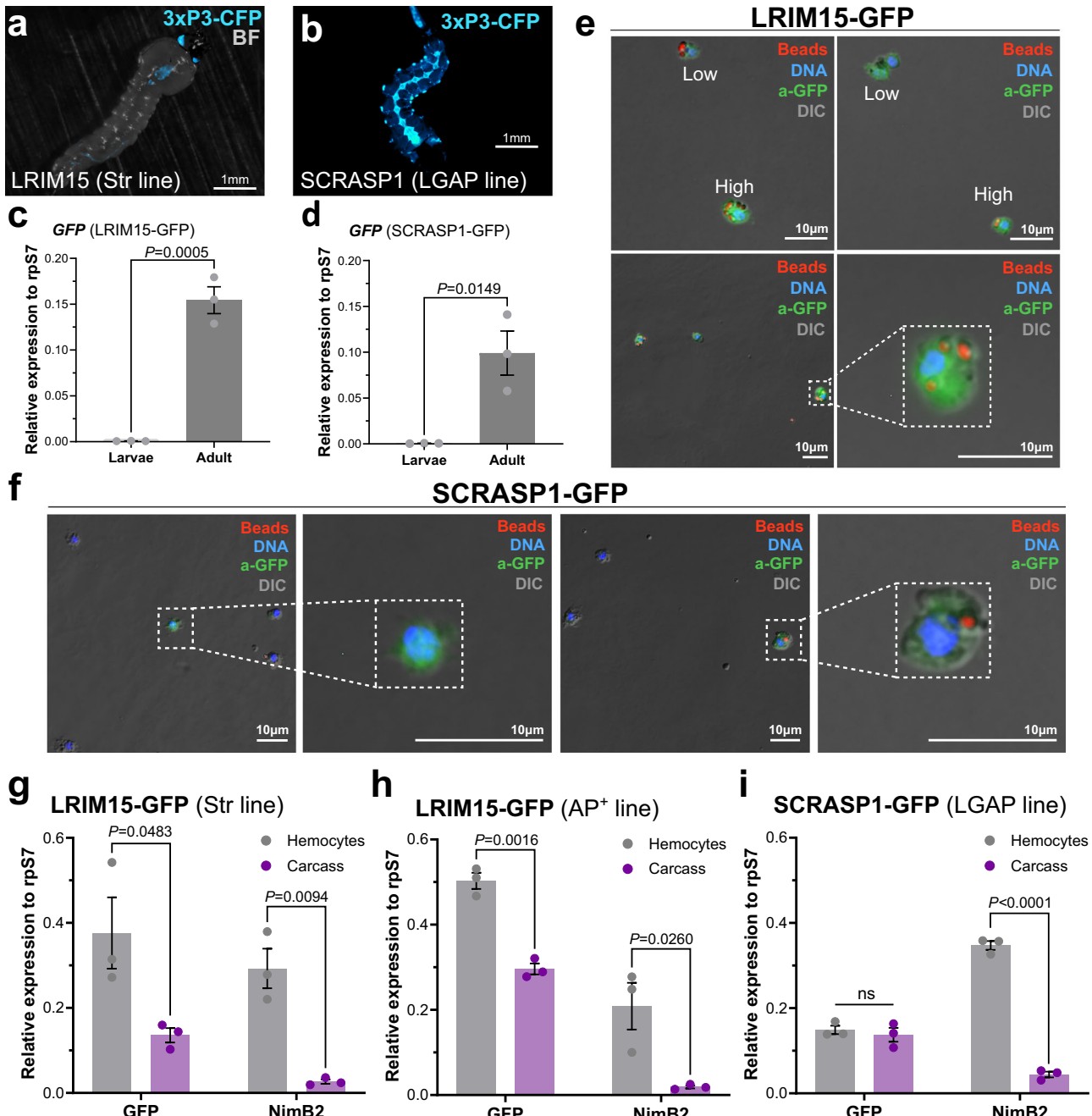

**Fig. 2 | Molecular characterization of the putative granulocyte markers, LRIM15 and SCRASP1.** GFP fluorescence was examined in whole mount fourth-instar larvae from LRIM15 (**a**) or SCRASP1 (**b**) transgenic lines. Scale bars: 1 mm. Potential differences in *GFP* expression between larvae and adult mosquitoes were examined by qPCR for both LRIM15 (**c**) or SCRASP1 (**d**) transgenic lines. Expression data are displayed as the mean ± SE from three independent replicates (*N* = 3, dots) of pooled (*n* = ~10) larvae or adult female mosquitoes, with values relative to rpS7 expression. Significance was determined using a two-tailed unpaired Student's *t*-test. Exact *P* values are displayed in the figure. Immunostaining of adult hemocytes using an antibody specific to GFP (a-GFP) reveals various GFP⁺ hemocyte populations with respect to fluorescence intensity and phagocytic capacity in the **e** LRIM15-GFP and **f** SCRASP1-GFP transgenic lines. Subpanels demonstrate different representative cell phenotypes. Scale bar: 10 μm. *GFP* expression was enriched in hemocyte populations as compared to carcass tissue in both LRIM15 lines (**g** and **h**), although no difference was observed in the SCRASP1 transgenic mosquitoes (**i**). For (**g**–**i**), the hemocyte-specific expression of *NimB2* was used as a positive control for gene expression analysis. Expression data are displayed as the mean ± SE from three or more independent replicates (*N* = 3, dots) of pooled hemolymph perfusions (*n* = >30 adult female mosquitoes) or adult female mosquitoes (*n* = ~10) following perfusion (carcass), with values relative to rpS7 expression. Significance was determined using multiple two-tailed unpaired *t*-tests and a Holm−Šídák correction. Adjusted *P* values are displayed in the figure where significant. ns not significant. Source data are provided as a Source Data file.

aforementioned genetic markers for PPO6⁺, SPARC⁺, and LRIM15⁺ immune cells, we now have the ability to examine mosquito immune cell populations at high resolution by combining analysis of cellular properties (DNA content, size, granularity) with morphological phenotypes (cell imaging).

To examine mosquito hemocyte populations by IFC, we first applied the use of this technology to characterize immune cells in wild-type *An. gambiae*. Using nuclear staining (DRAQ5) and real-time imaging, we gated *An. gambiae* hemocytes to select only for cells with clear morphology, whereas positive events for nuclear staining but

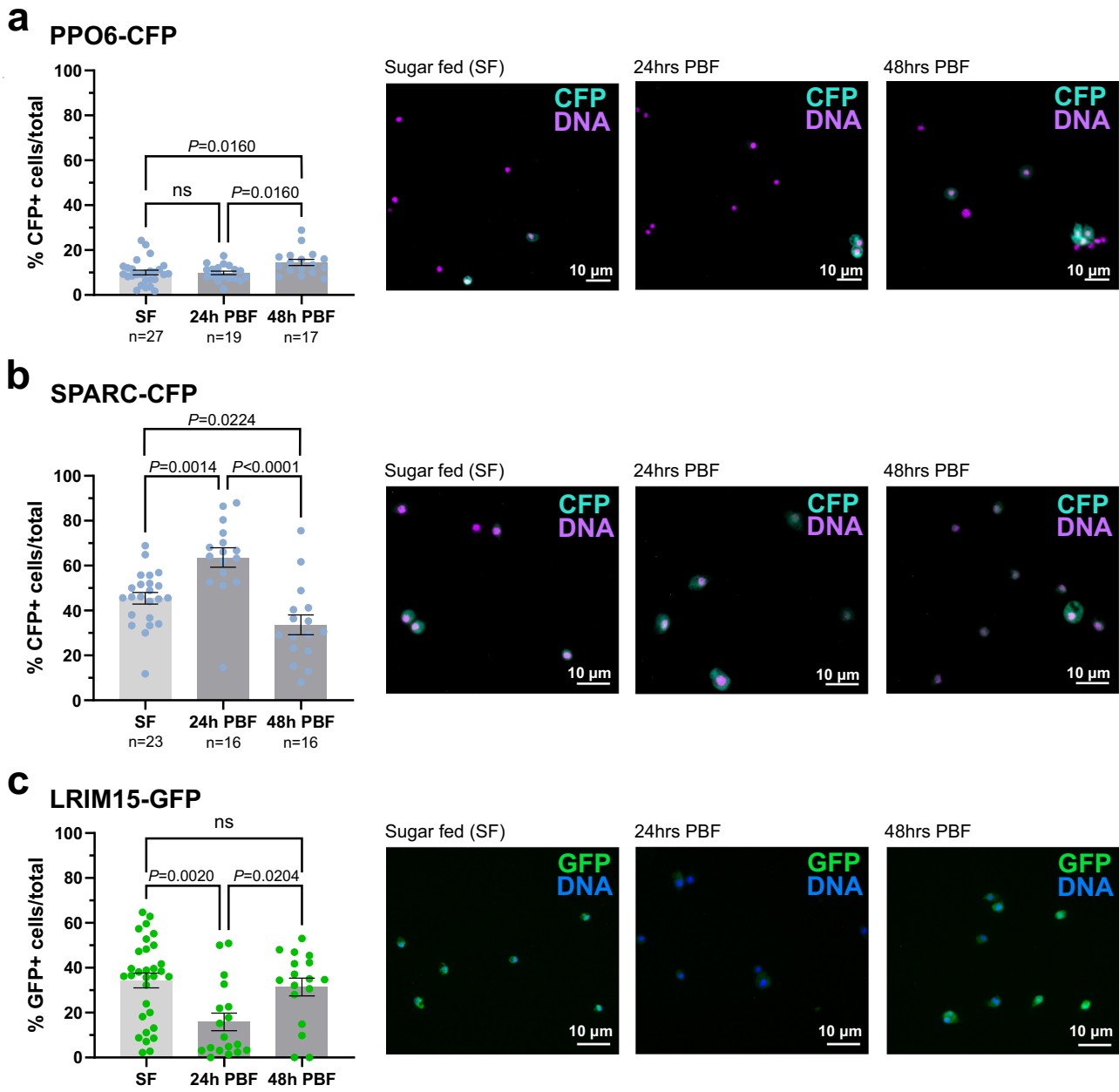

**Fig. 3 | Blood-feeding influences PPO6⁺, LRIM15⁺, and SPARC⁺ immune cell populations.** The percentage of PPO6⁺ (**a**), SPARC⁺ (**b**), and LRIM15⁺ (**c**) hemocytes were evaluated under sugar-fed (SF), at 24 h post-feeding (24 h PBF), and at 48 h post-feeding (48 PBF). For each experimental condition in (**a**–**c**), data display the percentage of CFP+ (**a**, **b**) or GFP+ cells (**c**) of the total hemocytes collected from individual mosquitoes (dots). *n* number of individual mosquitoes examined. Data from three independent biological replicates were pooled and displayed as the mean ± SE. Differences in the shading of the bar graphs highlight differences between sugar-fed (light gray) and blood-fed (dark gray) conditions. Statistical analysis was performed using a one-way ANOVA with a Holm–Šídák comparison test. Adjusted *P* values are displayed in the figure where significant. ns not significant. For each transgenic construct, representative images are displayed at right for each experimental condition. Scale bars represent 10 μm. Source data are provided as a Source Data file.

without discernible cellular morphology, were gated out as debris (Supplementary Fig. 10). Through this approach, mosquito hemocytes were most notably distinguished by DNA content or ploidy as previously[11,14,26], resulting in the identification of five distinct subpopulations based on DNA content, thus designated as P1-5 (Fig. 4a and Supplementary Fig. 10). Further examination of these P1-5 subpopulations using light loss (a measure reflecting the combined effects of cell size, granularity, and density) enabled the separation of cells with similar ploidy into additional subgroups, ultimately resulting in the characterization of 12 immune cell subtypes displaying distinct cell properties of size and ploidy (Fig. 4a and Supplementary Fig. 11). To better display the relationships between these immune cell subtypes, cells were visualized using UMAP and t-SNE (Fig. 4b). These results reveal clearly delineated cell clusters with minimal overlap, corroborating the efficiency of our gating strategy. Moreover, the clear separation of certain cell groups, including the P2.1, P4.1, P4.2, P5.1, and P5.2 groups (Fig. 4b), indicates the possibility of specialized functions for each of these immune cell subtypes. In contrast, the close spatial relationships observed for the remaining clusters potentially reflect their similar function and the possibility of phenotypic plasticity.

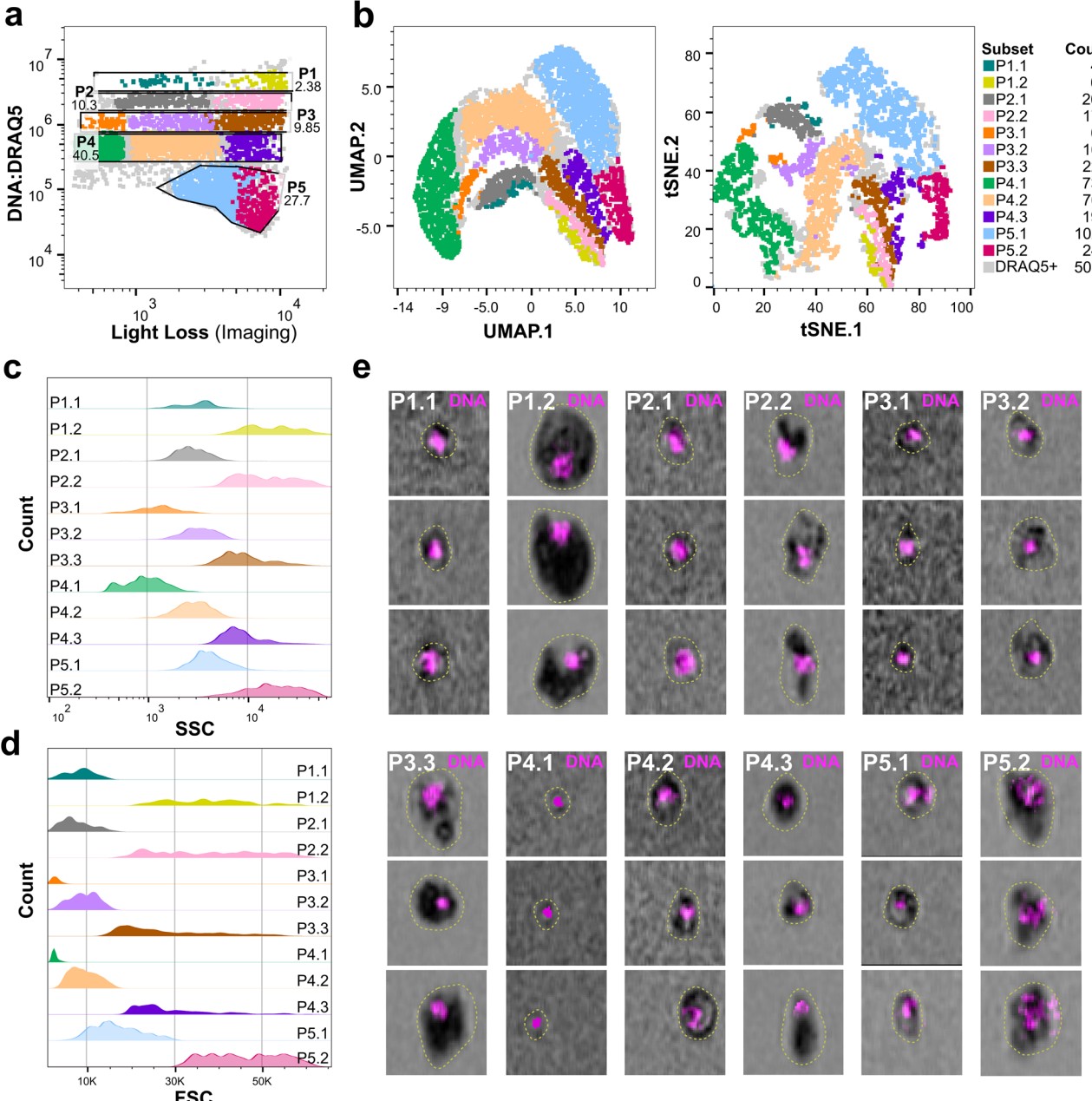

**Fig. 4 | Imaging flow cytometry reveals multiple hemocyte subpopulations.** The characterization of hemocytes using spectral imaging flow cytometry from naïve, wild-type adult females enables the primary classification of hemocytes based on DNA content (DRAQ5 signal; vertical *y*-axis), revealing five cell clusters (P), or tiers, based on DNA ploidy (**a**). When light loss measurements were incorporated (horizontal *x*-axis), additional subgroups were defined within each cell cluster (shaded by different colors). **b** Immune cell subtypes were analyzed by Uniform Manifold Approximation and Projection (UMAP) or t-Distributed Stochastic Neighbor Embedding (t-SNE) based on DRAQ5 signal, Maximum Intensity Forward Scatter (FSC), Maximum Intensity Side Scatter (SSC), and Maximum Intensity Light Loss characteristics, allowing for the clustering of hemocyte subpopulations according to relatedness. For each defined subtype, immune cell distributions are displayed for granularity (SSC; **c**) and size (FSC; **d**). **e** Representative images of each immune cell subcluster (P1.1–P5.2) are displayed, highlighting differences in size (outlined by dotted line), light loss, and DNA content (DRAQ5; magenta). UMAP analysis was performed using the Euclidean distance metric, and t-SNE was performed with opt-SNE learning configuration using FlowJo V10.10.0, including the following parameters: DRAQ5 signal, Maximum Intensity Forward Scatter (FSC), Maximum Intensity Side Scatter (SSC), and Maximum Intensity Light Loss. Graphs in (**a** and **b**) display a single replicate of three independent biological experiments. Source data are provided as a Source Data file.

Granularity is one of the most prominent features of phagocytic immune cells across metazoa and has served as a defining feature of mosquito granulocytes. Side scatter analysis (SSC), a measurement of granularity, established P3.1 and P4.1 clusters as the least granular cells, while the remaining clusters exhibited medium (clusters: P1.1, P2.1, P3.2, P4.2, and P5.1) or high (clusters: P1.2, P2.2, P3.3, P4.3, and P5.2) granularity (Fig. 4c). Forward scatter analysis (FSC), which allows for

discrimination of cells by size, revealed a proportional relationship between granularity and cellular size, where those cells displaying the least granularity were the smallest in size, while clusters with medium or high granularity ranged from medium to large size (Fig. 4d). Among these, the P3.1 and P4.1 groups displayed the smallest size with minimal variation, while P1.2, P2.2, P3.3, P4.3 and P5.2 groups exhibited the largest variation in size (Fig. 4d). Real-time imaging corroborated these

**Table 1 | Summary of gated hemocyte subpopulations**

| Population | Presumed subtype | % of cells[a] | FSC (M ± SE) | SSC (M ± SE) | DRAQ5 (M ± SE) | Phagocytic | % PPO6 + | % SPARC + | % LRIM15 + |
|---|---|---|---|---|---|---|---|---|---|
| P3.1 | Prohemocyte | 3.3 | 2486 ± 28 | 1011 ± 190 | 964,667 ± 35,951 | – | 0.0 | 0.2 | 0.0 |
| P4.1 | Prohemocyte | 27.7 | 2397 ± 71 | 781 ± 88 | 449,333 ± 15,344 | – | 0.5 | 0.0 | 0.0 |
| P1.1 | Oenocytoid | 2.0 | 7166 ± 650 | 2936 ± 298 | 4,866,667 ± 121,975 | – | 0.1 | 0.0 | 0.1 |
| P2.1 | Oenocytoid | 5.6 | 5201 ± 949 | 2255 ± 272 | 2,040,000 ± 60,000 | – | 0.1 | 0.1 | 0.3 |
| P3.2 | Intermediate | 3.2 | 8064 ± 528 | 2810 ± 288 | 978,833 ± 27,668 | + | 2.3 | 1.1 | 1.3 |
| P4.2 | Intermediate | 14.6 | 8300 ± 432 | 2852 ± 59 | 474,667 ± 18,800 | + | 4.2 | 7.1 | 4.9 |
| P1.2 | Granulocyte | 0.9 | 31,844 ± 4088 | 17,247 ± 2744 | 4,316,667 ± 156,241 | ++ | 4.8 | 1.4 | 2.9 |
| P2.2 | Granulocyte | 2.2 | 38,393 ± 528 | 12,392 ± 2417 | 2,356,667 ± 57,831 | ++ | 7.1 | 2.7 | 4.1 |
| P3.3 | Granulocyte | 3.0 | 24,387 ± 822 | 8212 ± 139 | 1,040,000 ± 30,000 | +++ | 12.7 | 6.2 | 9.9 |
| P4.3 | Granulocyte | 3.0 | 26,046 ± 1648 | 7889 ± 494 | 433,506 ± 51,132 | +++ | 17.6 | 13.7 | 12.5 |
| P5.1 | Granulocyte | 30.5 | 13,982 ± 1642 | 3623 ± 237 | 85,220 ± 1832 | +/++ | 39.9 | 61.2 | 58.3 |
| P5.2 | Granulocyte | 4.0 | 41,125 ± 2717 | 14,606 ± 1988 | 56,426 ± 1915 | ++/+++ | 10.7 | 6.3 | 5.7 |

[a]percentage of total immune cells gated into hemocyte subpopulations.
Forward Scatter (FSC), Side Scatter (SSC), and nuclear staining (DRAQ5) intensity are presented as mean ± SE values from three biological replicates.
Phagocytic capacity of hemocyte subtypes classified based on bead uptake rates as depicted in Fig. 6e.

observations regarding the ploidy, size, and granularity of each cell cluster, providing additional insights into the morphological features and capturing the highly structured immune cell landscape of *An. gambiae* (Fig. 4e).

These immune cell features are summarized in Table 1, where we provide information of each cell population regarding their presumed hemocyte subtypes, abundance, FSC, SSC, and DNA content using signal measurements for DRAQ5.

### Flow cytometry analysis of transgenic immune cell markers

Using our analysis of wild-type immune cell populations as a reference (Fig. 4), we next explored the properties of our SPARC+, PPO6+, and LRIM15+ cells to better understand the properties of the immune cells labeled by each genetic marker. Consistent with our microscopy analysis (Fig. 3), we see similar patterns of abundance for each transgenic construct, albeit at lower percentages of cells in our IFC analysis. Mosquitoes of the LRIM15-GFP line exhibited 15.2 ± 1.44% of GFP+ cells, SPARC-CFP displayed the highest proportion with 27 ± 0.87% of cells, and PPO6 labeled 8.7 ± 0.6% of cells (Supplementary Fig. 12 and Supplementary Table 3). Further analysis of fluorescent hemocytes in each transgenic line revealed their cellular composition (Fig. 5a–f). Hemocytes labeled by LRIM15-GFP were primarily composed of cells with medium to high granularity, with the P5.1 cluster accounting for 58.3% of the population (Fig. 5a, b and Table 1). SPARC-CFP hemocyte populations displayed strong similarity to LRIM15+ cells, with 61.2% of cells also belonging to the P5.1 cluster (Fig. 5c, d and Table 1). While PPO6-CFP fluorescent hemocytes were also predominantly localized to cluster P5.1, they comprised a larger proportion of P2.2, P3.3, and P4.3 cells (Fig. 5e, f and Table 1). Together, these results indicate that while there are differences in the abundance of LRIM15+, SPARC+, and PPO6+ cells, their cellular properties (i.e., granularity) suggest that each of these promoter constructs most likely label subpopulations of granulocytes in varying capacity.

Based on these similarities in the types of cells that are labeled with our respective LRIM15, SPARC, and PPO6 constructs, we wanted to examine the potential overlap between transgenic constructs. To address this, we outcrossed either SPARC-CFP or PPO6-CFP mosquitoes with LRIM15-GFP. Following the selection and establishment of mosquitoes with both fluorescent markers, we examined the presence/absence of CFP+ and GFP+ cells in these mixed genetic backgrounds. SPARC-CFP and LRIM15+ hemocytes displayed a moderate degree of overlap with ~30% of total fluorescent cells expressing both markers, with the majority of CFP+/GFP+ cells belonging to the P3.3 and

P4.3 clusters (Fig. 5g). In addition, while the P5.1 cluster individually represented the majority (>50%) of the SPARC-CFP+ or LRIM15-GFP+ individual cell populations, the P5.1 cluster is underrepresented in cells co-expressing both genetic markers (Fig. 5g). While there is some overlap between the PPO6 and LRIM15 markers (~10% of total fluorescent cell population), the two groups were clearly delineated, with the P5.1 cluster representing the bulk of CFP+/GFP+ hemocytes (Fig. 5h). Together, these data suggest that there are distinct differences in the cell populations that co-express LRIM15+/SPARC+ and LRIM15+/PPO6+ cells, underscoring the existence of distinct genetic features within morphologically similar immune cell populations that support a greater complexity of hemocyte marker expression in mosquito immune cell populations.

### Analysis of the phagocytic capacity of mosquito immune cell subtypes

In mosquitoes, granulocytes are the primary mediators of phagocytosis[45], yet recent efforts have defined additional complexity in mosquito granulocyte populations[11,29] and potential differences in their phagocytic capacity[19]. To more closely examine immune cells involved in phagocytosis, we again employed IFC using fluorescent beads to examine the phagocytic capacity of *An. gambiae* immune cells.

Given the high fluorescence intensity of beads used in our experiments, we slightly adjusted our initial gating strategy to eliminate the potential spillover of red fluorescence into the DRAQ5+ channel (Supplementary Fig. 13). The instrument's high resolution, coupled with imaging, allowed us to exclude cell debris or bead singlets from the analysis to focus exclusively on phagocytic cells with stained nuclei (Supplementary Fig. 13). When examined, phagocytic cells displayed high levels of light loss, suggesting that mosquito immune cells with greater density are more likely to engage in phagocytosis (Fig. 6a). Consistent with previous studies[19], hemocytes displayed varying levels of phagocytic capacity indicative of the number of beads taken up by the cell, allowing us to classify them into three subpopulations: low, medium, and high (Fig. 6b). When our immune cell classifications are distinguished as either non-phagocytic or phagocytic cells, we see that some cell populations (P3.1 and P4.1) completely lack the ability to undergo phagocytosis, while others vary in their phagocytic capacity from minimal events (P1.1 and P2.1) to those, such as P3.3 and P4.3, with very high capacity (Fig. 6c, d and Table 1). Together, cell populations undergoing phagocytosis were primarily represented by P1.2, P2.2, P3.3, P4.3, P5.1, and P5.2 cell

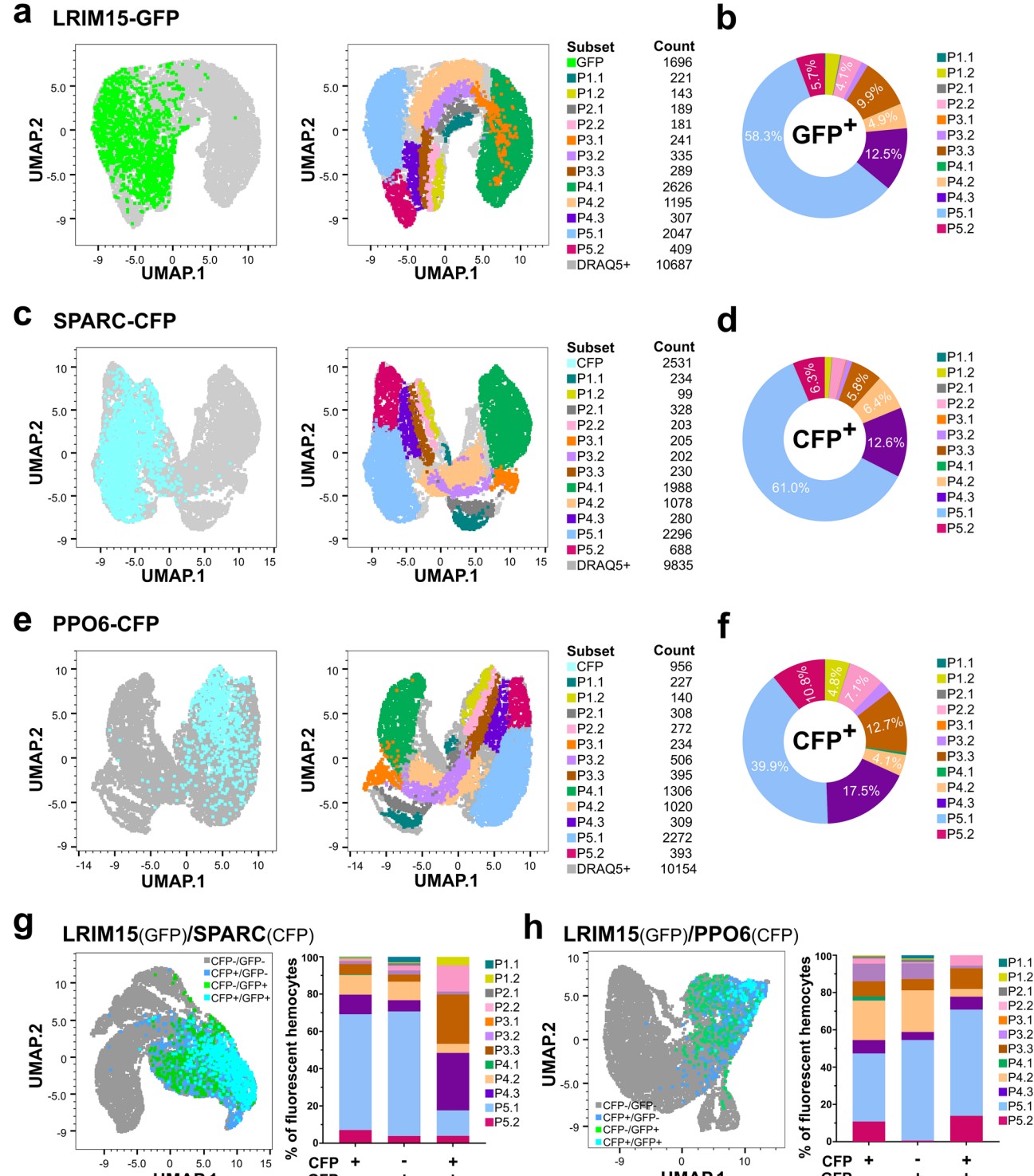

**Fig. 5 | Flow cytometry analysis of LRIM15⁺, SPARC⁺, and PPO6⁺ hemocyte markers.** Flow cytometry analysis was performed on immune cell populations resulting from LRIM15⁺, SPARC⁺, and PPO6⁺ individual transgenic lines (**a–f**). Representative UMAPs resulting from spectral imaging flow cytometry for LRIM15-GFP (**a**), SPARC-CFP (**c**), and PPO6-CFP transgenics (**e**) display the overall cell populations and the composition of fluorescent cells as an overlay using the established gating for each of the 12 hemocyte subpopulations identified in wild-type mosquitoes. The total number of events for each cell classification are displayed on the right of each figure subpanel. These distributions are summarized in pie charts to display fluorescent cell subtypes for LRIM15-GFP (**b**), SPARC-CFP (**d**), and PPO6-CFP transgenics (**f**). Data were averaged from three independent

biological replicates. To determine potential overlap between transgenic markers, crosses were performed to establish either LRIM15⁺/SPARC⁺ (**g**) or LRIM15⁺/PPO6⁺ (**h**) genetic backgrounds. For each genetic background, the abundance of GFP⁺, CFP⁺, and GFP⁺/CFP⁺ cells were examined by overlaying fluorescent cell populations on the UMAP and summarized in bar graphs to denote the hemocyte clusters represented for each fluorescent cell subtype. Data display the average of three independent biological replicates. UMAP analysis was performed using the Euclidean distance metric using FlowJo V10.10.0, including the following parameters: DRAQ5 signal, Maximum Intensity Forward Scatter (FSC), Maximum Intensity Side Scatter (SSC), and Maximum Intensity Light Loss. Source data are provided as a Source Data file.

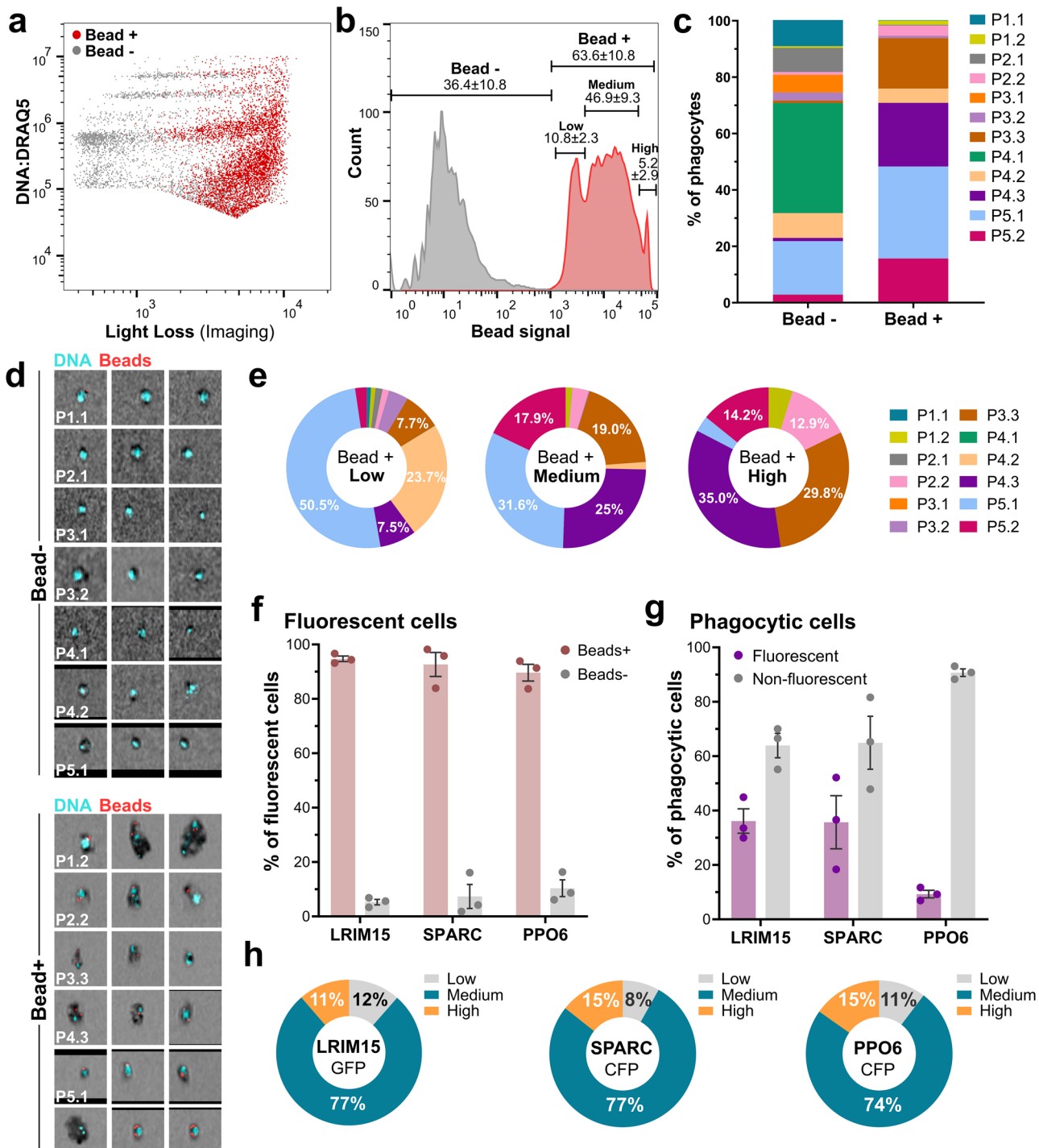

**Fig. 6 | Mosquito immune cells vary in their phagocytic capacity.** The injection of red fluorescent beads prior to perfusion enabled flow cytometry analysis of phagocytosis in mosquito immune cells (**a**) and the identification of multiple phagocytic immune cell phenotypes based on the intensity of bead signal (**b**). This resulted in the identification of non-phagocytic and phagocytic immune cell sub-types (**c**) that were confirmed by imaging (**d**). When phagocytic cells were further distinguished by bead signal intensity, we identified immune cell subtypes with low, medium, or high phagocytic capacity (**e**). Similar experiments with our LRIM15, SPARC, and PPO6 transgenic lines revealed the phagocytic ability of fluorescent immune cells (**f**) and the overall proportion of phagocytic immune cells (**g**), which are displayed as the mean ± SE of three biological replicates. **h** The phagocytic capacity of each transgenic line is visualized as the percentage of fluorescent cells displaying low, medium, or high bead-positive cells and displayed as the average of three independent biological replicates. Dot plots represent single replicates of three independent biological experiments. Source data are provided as a Source Data file.

clusters (Fig. 6c), which exhibit larger size and greater light loss sug-gestive of granulocytes (Fig. 6d). Of note, cell populations such as P3.2 and 4.2, exhibit much lower levels of phagocytic ability (Table 1), suggesting that they may represent intermediate or immature immune cell populations.

To gain further insight into the observed differences in phago-cytosis between immune cell clusters, we examined the composition of immune cells displaying low, medium, and high phagocytic capacity (Fig. 6e). Cells displaying "low" phagocytic capacity showed the high-est representation of immune cell subtypes, while fewer immune cell

clusters were represented in the "medium" and "high" phenotypes (Fig. 6e), suggesting that there is specialization of some immune cells to be more phagocytic. This is supported by the P5.1 cluster, which was enriched in cells with low and medium phagocytic capacity yet was underrepresented among cells displaying the highest bead uptake (Fig. 6e). Alternatively, these observations could also be partially justified by cell size, assuming a proportional relationship between cell size and phagocytic capacity. Consistent with this hypothesis, cells with higher phagocytic capacity (P1.2, P2.2, P3.3, P4.3, and P5.2 clusters) (Fig. 6e) also represent cells of the largest cell size and granularity among our immune cell subtypes (Fig. 4c, d and Table 1). Yet, cells displaying the highest phagocytic capacity (P3.3 and P4.3; Fig. 6e) are smaller in size (FSC) than other phagocytic cell types (Table 1), suggesting that these cells have become highly specialized in their phagocytic abilities.

When we performed similar analysis with our LRIM15, SPARC, and PPO6 transgenic lines (Supplementary Fig. 14), >90% of fluorescently-labeled hemocytes displayed phagocytic capacity (Fig. 6f), providing further support for our observations that these transgenic constructs predominantly labeled populations of granulocytes (Fig. 5 and Table 1). However, only a subset of phagocytic cells was labeled in each of our transgenic lines, with LRIM15 and SPARC labeling up to 36% of phagocytes, while PPO6[+] cells representing only ~10% of phagocytic cells (Fig. 6g). When further examined for potential differences in phagocytosis, each of the transgenic markers labeled cell populations primarily comprised of cells with medium phagocytic capacity (Fig. 6h). This is consistent with the phagocytic abilities of P4.3 and P5.1 (Fig. 6e) which are predominantly labeled with each of the transgenic constructs (Fig. 5).

## Discussion

Hemocytes are integral components of mosquito innate immunity, with essential roles in defining vector competence and disease transmission. While mosquito hemocyte populations have traditionally been subdivided into three subtypes based on their morphological properties, recent studies have suggested a more complex and dynamic composition of immune cell subtypes[11,29]. However, the lack of genetic tools for mosquito hemocytes has been a significant hurdle for further studies to unravel their complexity, thereby causing a reliance on morphological properties that have only confounded their function[22]. Herein, we describe the development of multiple hemocyte markers and their utility to provide an unbiased classification of *A. gambiae* immune cells using genetic and functional immunophenotyping.

With an initial goal to identify promoters that would comprehensively label all mosquito hemocyte populations or that specifically target granulocytes, we identified three functional promoters (PPO6, SPARC, and LRIM15) able to successfully drive the robust expression of fluorescent markers in *A. gambiae* hemocytes. Additional promoter constructs using putative regulatory regions for SCRASP1 and NimB2, respectively, displayed limited or no marker expression despite the prominence of both genes in previous mosquito hemocyte studies[11,19,27,28,42]. This suggests that additional regulatory regions are likely required for both promoters to drive high levels of heterologous expression in mosquito immune cell populations. This may be addressed through future experiments using similar transposon-based experiments with extended regulatory regions or through a knock-in approach to drive expression using the endogenous gene[46].

Based on previous single-cell studies[11,28] and immunofluorescence experiments[19,26], we had expected that the PPO6 and SPARC constructs could potentially be used to drive expression across hemocyte subtypes and serve as pan-hemocyte markers. This was supported by the previously described use of the PPO6 promoter to drive expression in *Anopheles* hemocytes[28,36]. Consistent with previous observations[28,36], the PPO6 promoter drove hemocyte expression in both larval and adult stages at comparable levels, displaying similar visible fluorescent phenotypes. However, we found that the activity of the promoter was limited to a small subset of immune cells, accounting for only ~10% hemocytes using hemocytometer and flow cytometry approaches, in agreement with previous insinuations that the PPO6 promoter was active in only a subset of granulocytes[35]. Additional immunofluorescence experiments performed herein support that the PPO6-CFP construct labels the majority of PPO6[+] immune cells, yet is incomplete in labeling all PPO6[+] cells.

Similar patterns of expression were observed for the SPARC-CFP construct in larvae and adults, although the proportions of SPARC[+] hemocytes were significantly higher, reaching ~45% in microscopy experiments and ~27% of the total population when measured via flow cytometry. However, these numbers fall short of achieving a "universal" or "panhemocyte" promoter that would match the previous descriptions of PPO6 and SPARC expression in prior single-cell studies[11,28,29]. At present, it is unclear if these observations for the PPO6 and SPARC promoter constructs are due to the use of inadequate regulatory regions needed in our promoter constructs or whether additional unknown mechanisms of post-transcriptional regulation influence their expression across immune cell subtypes.

While SPARC expression is enriched in hemocytes, we also detected CFP expression in the fat body, which aligns with the tissue specificity of the *Drosophila* SPARC ortholog. Previous studies have shown that SPARC is localized in *Drosophila* hemocytes[47] and fat body cells with a unique role in regulating the polymerization and deposition of collagen IV to sustain basal membrane integrity and fat body homeostasis[48–50]. Based on these similar expression patterns, this suggests that SPARC[+] hemocytes may be involved in maintaining tissue homeostasis and production of the basal lamina, potentially expanding the utility of the promoter beyond hemocyte function to other aspects of mosquito physiology.

With previous studies implicating the expression of LRIM15 in phagocytic immune cell populations[11,27], the LRIM15 promoter construct performed as expected, driving strong fluorescent marker expression in mosquito granulocytes. However, we observed *GFP* expression and LRIM15[+] cells only in adult mosquitoes, suggesting that mosquito hemocytes undergo additional hematopoiesis or maturation shortly after adult eclosion. This is supported by previous studies highlighting differences between larval and adult immune responses, which includes increased phagocytic activity in adult mosquitoes[51]. While speculative, this suggests that the adult expression of LRIM15, and potentially other granulocyte-specific markers, may account for these immunological differences between mosquito life stages. Moreover, with only a limited understanding of larval hemocytes, these observations highlight the need for future studies to compare mosquito immune cell populations across development to better understand hematopoiesis and the physiological stimuli that may influence immune cell maturation.

Evidence suggests that mosquito hemocyte populations display plasticity and undergo significant alterations in response to physiological signals such as blood-feeding[14,26]. Our data provide initial proof-of-principle experiments supporting that PPO6[+], SPARC[+], and LRIM15[+] cells are dynamic in their abundance. Specifically, we observe an increase in circulating PPO6[+] cells at 48 h post-blood feeding, with this change specifically attributed to the increased abundance of PPO6[low] immune cell populations. We also observe an increase in SPARC[+] cells at 24 h post-blood meal, yet by 48 h post-feeding, there is a significant reduction in their abundance. In contrast, the abundance of LRIM15[+] cells decreased at 24 h post-blood meal and returned to normal levels by 48 h post-feeding. While these observations generally align with previous findings indicating the transient activation of mosquito hemocytes[26], information as to how the physiological effects resulting from blood-feeding, or other potential stimuli, influence these mosquito immune cell populations remains elusive. Given that PPO6[+],

SPARC[+], and LRIM15[+] cells collectively display characteristics of granulocyte populations, with only partial co-localization between these cell markers in flow cytometry experiments, our data suggest that there is greater complexity in mosquito granulocyte populations that may reflect different levels of maturation, activation, or specialized immune function.

Using our PPO6[+], SPARC[+], and LRIM15[+] transgenic lines, we employed recent advances in IFC to characterize *An. gambiae* hemocyte populations by ploidy, size, granularity, morphology, and phagocytic capacity. Consistent with previous studies[11,14,26], our data demonstrate that mosquito hemocytes are readily distinguished by differences in DNA content or ploidy. Using DRAQ5 to provide indirect measures of DNA content, our data demonstrate that each hemocyte subpopulation has at least two different levels of DNA, suggesting an approximate doubling (or further expansion) of DNA content among similar immune cell subtypes. While this may encompass some cells undergoing normal mitosis and cell division, the large proportion of immune cells displaying polyploidy is suggestive that endocycling (endomitosis or endoreplication) is an integral aspect of mosquito hemocyte biology. Cell polyploidy is common in insects and has been implicated in a variety of biological functions to increase transcriptional activity and protein secretion[52,53]. With previous studies in mosquito cell lines suggesting that endoreplication occurs in response to pathogen infection and is essential for immune priming[54,55], polyploidy could represent a unique methodology used by mosquito immune cells for specialized immune functions or to enhance the response time to pathogen challenge. However, one limitation of our approach is that the DNA levels needed to distinguish between diploid, tetraploid, or other polyploid cells remains unknown. As a result, further studies are required to fully determine the effects of ploidy on immune cell function.

When DNA contents is paired with traditional measurements of size and granularity (which are routinely used as a proxy of determining cell function[56,57]), we identify a total of twelve immune cell subpopulations in *An. gambiae*. Among these cell types, we see a clear delineation of non-phagocytic and phagocytic cells, which are readily distinguished by differences in size and granularity. Utilizing our IFC methodology's ability to both visualize and physically characterize these cell types, we believe that the non-phagocytic cell types represent prohemocyte (clusters P3.1 and P4.1) and oenocytoid (P1.1 and P2.1) cell populations. In contrast, phagocytic immune cell populations are reminiscent of granulocytes (such as P1.2, P2.2, P3.3, P4.3, P5.1, and P5.2), with the potential that one or more of these cell clusters may correspond to the megacyte lineage[29,41]. Additional populations observed in P3.2 and P4.2 may represent intermediate or immature immune cell populations based on differences in size, granularity, and limited phagocytic ability, potentially representing the immature granulocyte populations defined in previous single-cell studies[11].

Based on the morphology and phagocytic properties of PPO6[+], SPARC[+], and LRIM15[+] cells, our data suggest that each of the aforementioned markers predominantly label granulocyte populations. Although the abundance of PPO6[+], SPARC[+], and LRIM15[+] cells is similar between granulocyte subtypes, our co-expression studies resulting from combined crosses with either LRIM15-SPARC or LRIM15-PPO6 suggest that only a subset of granulocytes display both markers. This implies that there is additional complexity among these phagocytic cells, suggesting that potential differences in immune maturation or specialized cell function could further influence the expression of PPO6[+], SPARC[+], and LRIM15[+] cells.

Despite the significant advances in mosquito immune cell biology outlined in our study, several limitations and challenges remain. While our use of IFC is a strength of our approach, we anticipate further refinement of this methodology in its application for mosquito immune cells. This includes optimizing the conservative gating strategies employed in our study, which may have inadvertently included

or excluded specific immune cell subtypes. A key challenge remains in the identification of additional immune cell markers, where further discoveries would not only augment our current genetic tools, but also improve the resolution of mosquito immune cell characterization. For example, the inclusion of additional genetic markers that clearly delineate prohemocyte, oenocytoid, or specific granulocyte populations would further enhance our ability to resolve immune cell subtypes. This would enable additional precision and reproducibility in studies evaluating immune cell dynamics in response to physiological stimuli or infection, providing further refinement of mosquito immune cell subtypes and their contributions to mosquito innate immune function.

In summary, our study provides an essential foundation for future studies of mosquito immune cell biology where technical limitations have previously hindered progress. This includes the development of genetic resources that enhance the visualization of hemocyte subtypes and demonstrate the application of IFC technologies in an insect system, offering an unprecedented resolution of mosquito immune cells. These significant advancements now pave the way for opportunities to address fundamental questions in mosquito hemocyte biology, such as hematopoiesis, cell differentiation, immune plasticity, and the cellular responses to a variety of physiological stimuli (blood-feeding, infection, etc.) through reproducible methodologies. Moreover, the enhanced characterization of mosquito immune cells creates new opportunities for comparative immunology, offering methods to examine immune cell plasticity and conserved or divergent trajectories of immune cell populations across invertebrate and vertebrate species. As a result, we believe these findings provide a critical resource for future studies to expand our knowledge of mosquito immune cells and their contributions to mosquito vector competence.

## Methods

### Ethics oversight
All mosquito experiments were performed under the approval of Iowa State University's Institutional Biosafety Committee (IBC) protocols #24-085 and #24-086. No vertebrate animals were used in this study.

### Mosquito rearing
Transgenic and wild-type *Anopheles gambiae* mosquitoes (Keele strain[58]) were reared at 27 °C and 80% relative humidity, with a 14:10 h light: dark photoperiod cycle. Larvae were fed on commercialized fish flakes (TetraMin Tropical Flakes, Tetra), while adults were maintained on a 10% sucrose solution and fed on commercial sheep blood (HemoStat, #DSB050) for egg production.

### Mosquito embryo transformation
Transgenic mosquitoes were generated using the piggyBac transposon system. *Anopheles gambiae* (Keele) preblastoderm embryos were injected by the Insect Transformation Facility at the University of Maryland Institute for Bioscience & Biotechnology Research. All injections were performed using an injection solution containing 150 ng/μL of piggyBac vector and 175 ng/μL of hyperactive piggyBac transposase mRNA[59–61] under halocarbon oil as previously described[62]. After injections, the hatched insects that survived to adulthood were pooled based on sex and crossed with the wild-type strain *An. gambiae* (Keele). Progenies were screened for the expression of ECFP or DsRed integration markers at late larval stages. Individual transgenic lines were identified by distinct expression patterns of the ECFP or DsRed integration markers, which were used to establish unique colonies.

### Hemocyte-specific transgenic mosquitoes
Hemocyte-specific *An. gambiae* reporter lines were generated by fusing the promoter sequences that drive universal hemocyte or granulocyte-specific gene expression to the fluorescent markers CFP and GFP, respectively. Using previous transcriptomic, proteomic, and

functional data[11,27–29,36], the genes NimB2 (AGAP029054), SPARC (AGAP000305), and PPO6 (AGAP004977) were selected as putative universal hemocyte markers, while LRIM15 (AGAP007045) and SCRASP1 (AGAP005625) were chosen as granulocyte-specific markers. Promoter sequences encompassing the 5′ untranslated regions and up to ~2 kb upstream of the Transcription Start Site (TSS) of each gene, were downloaded from Vectorbase and were either PCR amplified or underwent de novo synthesis (Integrated DNA Technologies) (Supplementary Table 1). All primer pairs used for PCR amplification of the putative promoters are listed in Supplementary Table 4. Following amplification, PCR products were initially subcloned in pJET1.2/blunt (CloneJet PCR Cloning kit, Thermo Fisher, # K1232) for sequence verification by Sanger sequencing (DNA Facility, Iowa State University) prior to cloning into the respective piggyBac constructs.

### Genomic DNA extraction
Genomic DNA was extracted from pools of ten adult mosquitoes as previously[63,64] by homogenizing in Bender buffer (0.1 M NaCl, 0.2 M Sucrose, 0.1 M Tris-HCl, 0.05 M EDTA pH 9.1, and 0.5% SDS), followed by incubation at 65 °C for 1 h. After adding 15 μl of 8 M potassium acetate, samples were incubated for 45 min on ice and centrifuged for 10 min at maximum speed. Genomic DNA was ethanol-precipitated and resuspended in nuclease-free water.

### Plasmid construction
The open reading frame (ORF) of DsRed was excised from piggyBac-3 × P3-DsRed[65] with NcoI-NotI and replaced with either the ECFP ORF from piggyBac-ECFP-15 × QUAS_TATA-mcd8-GFP-SV40[66] (Addgene, #104878) or GFP ORF amplified from an existing pJET1.2-T7-GFP plasmid[67] with primers GFP-F-NcoI and GFP-R-NotI. Candidate hemocyte promoters were amplified with Phusion polymerase (Thermo Fisher Scientific, #F530S) using primers with AscI or FseI-AsiSI restriction sites, respectively, attached to the 5′-end of the forward or reverse primers (Supplementary Table 4) and cloned into the AscI and FseI restriction sites of the desired piggyBac plasmid. The ORFs of CFP and GFP followed by SV40 termination sequence were inserted at the 3′ end of each candidate promoter using the restriction sites AsiSI and FseI. All plasmid sequences were confirmed by Sanger sequencing prior to microinjection, with sequences of each construct provided in Supplementary Table 1.

### Mapping hemocyte-specific transgene insertion in Anopheles genome
To identify the integration sites of each hemocyte-specific transgene, we performed splinkerette PCR (spPCR) on the genomic DNA of each transgenic line as previously described[68,69]. Genomic DNA was extracted from pooled adult mosquitoes and digested with BglII or MspI for four hours. Splinkerette double-stranded oligos were synthesized to complement the sticky ends generated by BglII or MspI. Digested genomic DNA was ligated to the respective annealed splinkerette oligos with T4 DNA ligase (Thermo Fisher Scientific) at 4 °C overnight. PCR reactions were performed using Phusion polymerase (New England Biolabs, #M0530S) as previously described[69]. A list of all primers used for spPCR is summarized in Supplementary Table 5. PCR fragments were gel purified using Zymoclean Gel DNA Recovery Kit (Zymo Research, #D4002) and cloned to pJET1.2/Blunt vector (CloneJet PCR Cloning kit, Thermo Fisher, #K1232) for Sanger sequencing. The recovered DNA sequences were mapped to the An. gambiae PEST reference genome using the blastn function in VectorBase.

### RNA extraction and gene expression analyses
Total RNA was extracted from whole mosquito samples (~10 adult female mosquitoes) using TriZol Reagent (Invitrogen, #15596018). RNA samples prepared from perfused hemolymph samples were isolated using the Direct-Zol RNA miniprep kit (Zymo Research, #R2052).

Two micrograms of whole mosquito-derived or 200 ng of hemolymph-derived total RNA were used for first-strand synthesis with the Luna-Script RT SuperScript Kit (New England Bioloabs, #E3010L). Gene expression analysis was performed with quantitative real-time PCR (qPCR) using PowerUp SYBRGreen Master Mix (Thermo Fisher Scientific, #A25742). qPCR results were calculated using the $2^{-\Delta Ct}$ formula and normalized by subtracting the Ct values of the target genes from the Ct values of the internal reference, rpS7[70]. All primers used for gene expression analyses are listed in Supplementary Table 6.

### Transgene expression in response to blood-feeding
To determine the effects of blood-feeding on transgenic lines, adult transgenic mosquitoes (3–5 days old) were allowed to feed on defibrinated sheep blood for 5 min using an artificial membrane feeder. At 24 h post-feeding, a minimum of 10 engorged female mosquitoes were separated from unfed and used for RNA extraction and gene expression analysis. All blood-feeding experiments were repeated at least three times.

### Hemolymph perfusion and fluorescent microscopy
Mosquito adult hemolymph was collected by perfusion using an anticoagulant buffer of 60% v/v Schneider's Insect medium, 10% v/v Fetal Bovine Serum, and 30% v/v citrate buffer (98 mM NaOH, 186 mM NaCl, 1.7 mM EDTA, and 41 mM citric acid; buffer pH 4.5) as previously described[15,18,19]. For perfusions, mosquitoes were perforated on the posterior abdomen and injected with anticoagulant buffer (~10 μl) into the thorax. Hemolymph samples from individual mosquitoes were placed on multi-test microscopic slides (MP Biomedicals, #096041805) and observed under a fluorescent microscope (Zeiss Axio Imager).

### Mosquito injections with clodronate liposomes
To determine the effects of phagocyte depletion on the activity of our promoter constructs, 3–5 days old transgenic mosquitoes were intrathoracically injected with control or clodronate liposomes as previously described[19,21,71]. At 24 h post-injection, total RNA was isolated from a minimum of 10 whole female mosquitoes and used for gene expression analysis by qPCR. Hemocyte ablation experiments were performed in at least three independent experiments.

### Immunostaining of mosquito hemocytes
Hemolymph was perfused from blood- or sugar-fed female mosquitoes at 24 h or 48 h post-blood meal and placed on multi-test microscopic slides. Hemocytes were allowed to adhere for 20 min and fixed with 4% PFA for 15 min at room temperature. Samples were blocked with 2% BSA and 0.1% TritonX-100 in 1× PBS at 4 °C overnight. The next day, samples were incubated overnight at 4 °C with mouse anti-GFP (Developmental Studies Hybridoma Bank, #DHSB-GFP-12A6), diluted by 1:50 in a blocking medium. The following day, cells were washed three times with 1× PBS and incubated with goat anti-mouse Alexa Fluor 488 antibody (Thermo Fisher Scientific, #A-11001) diluted by 1:500 in blocking medium for 1 h at room temperature. After five washing steps, samples were mounted with DAPI antifade medium and immediately examined under a fluorescent microscope (Zeiss Axio Imager).

Similar methods were performed to examine the protein co-localization of CFP and PPO6 in the PPO6-CFP transgenic line. Hemolymph was perfused from naïve adult female mosquitoes of the wild-type Keele strain and the PPO6 transgenic line, and allowed to adhere to multi-well microscope slides. Hemocytes were fixed and blocked as described above. For immunostaining, samples were co-incubated overnight at 4 °C with a mouse monoclonal anti-cyan fluorescent protein (CFP) antibody (Biosensis, #M-1300-100) and a rabbit anti-PPO6 antibody[19], diluted at 1:250 and 1:500, respectively, in blocking buffer. After incubation, samples were washed three times with 1× PBS

and subsequently incubated for 1 h at room temperature with Alexa Fluor-conjugated secondary antibodies: goat anti-mouse Alexa Fluor 488 (Thermo Fisher Scientific, #A-11001) and goat anti-rabbit Alexa Fluor 568 (Thermo Fisher Scientific, #A-11011), both diluted 1:500 in blocking buffer. Following five additional washes with PBS, samples were mounted using DAPI-containing antifade mounting medium and immediately examined under a fluorescence microscope (Zeiss Axio Imager). Hemocytes were screened for the presence or absence of CFP and PPO6 signals. The proportions of CFP$^{+/-}$ and PPO6$^{+/-}$ hemocytes were quantified by analyzing approximately 50 randomly selected cells per individual mosquito.

## Flow cytometry

To analyze wild-type and transgenic mosquito hemocyte populations, we performed IFC using the BD FACSDiscover S8 Cell sorter (BD Biosciences). To visualize the proportions of phagocytic immune cells, mosquitoes were injected with red fluorescent carboxylate-modified microspheres (Thermo Fisher Scientific, #F8821) at a final concentration of 2% (v/v) and allowed to recover for 30 min at 27 °C. Hemolymph was perfused from ~40 individual mosquitoes with an anticoagulant buffer in microcentrifuge tubes kept on ice, as previously described[19]. Samples were centrifuged at $2000 \times g$ at 4 °C for 5 min, the supernatant was discarded, and pellets were resuspended in 1 ml of ice-cold $1 \times$ PBS. Immune cell nuclei were counterstained with DRAQ5 (1:1000, BD Biosciences, # 564902) for 1 h on ice. After incubation, cells were washed once with $1 \times$ PBS to remove the excess stain and cell suspensions were transferred to 5 ml flow cytometry tubes.

Gating strategies were optimized using unstained wild-type hemocytes (to set thresholds for DRAQ5, GFP, and CFP) and bead-only controls (to exclude free beads). For bead-uptake assays, gating was refined to eliminate fluorescent bead singlets whose signal overlapped with DRAQ5. DRAQ5+ cells were then replotted based on bead signal intensity and cell counts, allowing reliable separation of bead$^-$ and bead$^+$ hemocytes, with the latter further divided into low, medium, and high uptake groups based on peak signal intensity. Initial gating of DRAQ5-stained hemocytes was performed in real-time during acquisition and aided by imaging to exclude debris. Five parental populations (P1–P5) were defined based on DRAQ5 intensity (DNA content), and further analyzed for size (FSC) and light loss intensity to distinguish additional morphologically distinct subpopulations. For each cell subpopulation, measurements of FSC, SSC, and DRAQ5 were determined as the mean ± standard error (SEM) of all cell in that subpopulation from three independent biological replicates.

To visualize and interpret the diversity of hemocyte subtypes, we applied dimensionality reduction techniques (t-SNE and UMAP) using a combination of size, granularity, DNA content (ploidy), and light loss parameters. A similar multidimensional workflow was also applied to transgenic mosquito lines (SPARC-CFP, PPO6-CFP, and LRIM15-GFP) for comparative analysis.

To examine fluorescently-labeled cell populations resulting from our transgenic promoter constructs, unstained wild-type cells were used to set baseline thresholds for GFP and CFP fluorescence, which were consistently applied across all samples, including bead-uptake assays. In experiments involving F1 progeny from SPARC-CFP or PPO6-CFP lines outcrossed with LRIM15-GFP, hemocytes were first gated based on ploidy (DRAQ5 intensity), and within this population, cells were analyzed based on CFP and GFP fluorescence intensity using the established wild-type thresholds. To assess the composition of hemocyte subtypes within transgenic reporter populations, we applied the predefined gating strategy (developed from wild-type populations) to the transgenic samples.

Flow cytometry analysis of wild-type or transgenic hemocytes with or without fluorescent beads was performed in three independent biological replicates. Data were analyzed with FlowJo v10.10.0 software.

## Software

Graphical data visualization was performed using GraphPad Prism (version 10.4.2). Microscopy images were initially captured and processed using ZEN and Zen lite software (Zeiss) prior to final processing with Adobe Photoshop (version 26.7). Flow cytometry analysis was performed with FlowJo software (version 10.10.0). Figures were created using Inkscape (version 1.0.2-2) and in some instances were supplemented using images developed with BioRender (BioRender 2024).

## Reporting summary

Further information on research design is available in the Nature Portfolio Reporting Summary linked to this article.

## Data availability

All data are included in the Supplementary Information or available from the authors, as are unique reagents used in this Article. The raw numbers for charts and graphs are available in the Source Data file whenever possible. Data files for flow cytometry experiments are provided using Iowa State University's open data repository, Data-Share (https://doi.org/10.25380/iastate.30192754). Plasmids used for genetic transformation and subsequent transgenic lines described in the Article are available upon request. Source data are provided with this paper.

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

## Acknowledgements

The authors would like to thank Emma Howell and David Hall for their assistance with mosquito maintenance, as well as Robert Harrell of the University of Maryland Insect Transformation Facility for his assistance with *An. gambiae* transgenesis. The piggyBac-3 × P3-DsRed plasmid was kindly provided by Peter Atkinson. This work was supported by R21AI166857, R01AI177540, and R01AI182256 to R.C.S. from the National Institutes of Health, National Institute of Allergy and Infectious Diseases.

## Author contributions

G.-R.S. and R.C.S. conceived the study. G.-R.S. and H.K. performed experiments. R.C.S. provided supervision and experimental oversight. G.-R.S. and R.C.S. wrote the initial draft of the manuscript. All authors reviewed and edited the final manuscript.

## Competing interests

The authors declare no competing interests.
