## [Transparent Peer Review file · Nature Communications]

Characterization of *Anopheles gambiae* immune cells through genetic and functional immunophenotyping

Corresponding Author: Dr Ryan Smith

Version 0:

Reviewer comments:

Reviewer #2

(Remarks to the Author)

This manuscript examines hemocyte populations considering morphology, ploidy, phagocytic capacity, and expression of hemocyte-specific genetic markers. The authors introduce the regulatory region of genes that were either expressed in all hemocytes (NimB2, PPO6 and SPARC) and some with granulocyte-specific expression (LRIM15 and SCRASP1) to drive expression of reporter genes (CFP or GFP) in specific hemocyte populations. The establishment of *Anopheles gambiae* transgenic mosquitoes that express reporter genes in specific hemocyte populations and the analysis of DNA content in hemocytes are important contributions. However, the transgenic lines had limited success, as two transgenic lines (NimB2-CFP and SCRASP1) had very low expression and could not be used. PPO6-CFP and SPARC-CFP are not “panspecific” hemocyte markers, as the fluorescence in adult females was mostly detected in granulocytes. Furthermore, although immunostainings with the LRIM15-GFP line detected phagocytic cells with low or high levels of GFP protein expression in adult females and the enrichment in mRNA levels between hemocytes and carcass was modest. The manuscript is well written, but it is complex and difficult to follow. A summary table indicating the expression level, hemocyte populations where the marker was expressed, the level of enrichment in hemocytes, the percentage of cells that express the marker, etc. would be very useful.

Major Points:

1. The weakest point of the manuscript is the limited integration of the classification of hemocyte subpopulations defined by DNA ploidy and morphology with the information obtained from the fluorescent markers and the three major functional classes of hemocytes known to be present in adult females (prohemocytes, oenocytoids and granulocytes). The authors state: “Our results comprehensively map the composition of mosquito hemocytes, classifying them into twelve distinct populations based on size, granularity, ploidy, phagocytic capacity, and the expression of PPO6, SPARC, and LRIM15 genetic markers.” However, the transgenic lines were not used to define the twelve populations and it is not clear how the reporter genes can help to distinguish the subpopulations. They define 12 subpopulations of hemocytes based on flow cytometry analysis of ploidy, size and granularity, but it is not clear what type of cell each of those subpopulations represent. Are they prohemocytes, oenocytoids or granulocytes? What is the functional importance of that classification?

Furthermore, the fluorescent proteins are only expressed in a low proportion of cell in the flow cytometry experiments (LRIM15, 15%; SPARC, 27 %; PPO6, 8.7%). In theory, if LRIM15 and SPARC are both granulocyte markers, so they should stain a similar proportion of cells. Furthermore, PPO6 was selected as a “panhemocyte” marker, but only stains a very small proportion of cells (8.7%). The low detection level could also result from sensitivity of the flow assays to detect fluorescence in smaller cells (prohemocytes and oenocytoids).

2. It will be informative to correlate the levels of fluorescent protein and the target protein itself. For example, the authors could do a double immunofluorescence staining of the PPO6-CFP line with an anti-CFP antibody together with an antibody to detect endogenous PPO6 expression. This could help to elucidate whether the promoter region that was chosen does not drive expression of the reporter that matches that of the endogenous gene. It is also possible that the mRNA expression used to select the markers does not match protein expression. For example, the mRNA could be present in all hemocytes, but only be transcribed in a specific subpopulation. Alternatively, the gene that was selected may have a general low level of expression and can only be detected in cells with higher ploidy.

3. The five levels of DNA ploidy that the authors identified by flow cytometry is very interesting. However, the methods section for the flow cytometry is not very detailed, and it is hard to understand how some of the assays were performed. Specially it is not clear how flow cytometry was used to detect cell ploidy. What were the standards used to determine cell ploidy? The presence of cells of with different sizes (small, medium and large) for P3 and P4 and (small and large) for P1, P2 and P5 hemocytes suggests that DNA replication takes place in the smaller cells, presumably prohemocytes, and precedes hemocyte differentiation. Are P3.1 prohemocytes that give rise to P3.3 phagocytic granulocytes, with P3.2 being an intermediate stage of differentiation??

4. LRIM15 and SPARC are only expressed in cells with higher ploidy and in larger cells (P5.1 P5.2, P4.3 and P3.3). Does this reflect low sensitivity of the fluorescence detection by flow cytometry, so that only cells with multiple copies of the genome and/or that are larger express enough protein to be detected?

Minor points:

1. Using the single cell analysis to choose universal markers might not be the best strategy considering that expression is inflated and relative to the level of expression in all other cell in the experiment. For example, if a gene has relatively low counts in a cluster of cells and no expression on the rest of the cells, it will have a high fold change, but the positive cells express low levels of that gene, that is highly specific. It raises the question if the markers were chosen based on expression level or specificity.
2. It is very hard to see the fluorescence signal in larvae when it is shown in "blue" (Fig. 1a& 1b), please modify the image to make hemocytes more visible by either making them green or cyan.
3. Figures 1g-h and 2g-l measure the enrichment of CFP and GFP in the hemocytes of transgenic lines by qPCR but the interpretation might be misleading. The enrichment does not mean that all hemocytes express the fluorescent marker but could reflect a small portion of hemocytes expressing high amounts of the mRNA of the fluorescent protein. This is further confirmed by microscopy after calculating the percentage of cells that express CFP.
4. Considering the percentage of cells that express the fluorescent protein in all 3 mosquito lines, indicates that expression is happening in a subset of hemocytes instead of all.
5. Low and high signals are hard to see in figures 2E. Please remove the DIC background placing an additional column in the panel.
6. Some figure legends lack crucial information regarding the figures. For instance: Figure 5K and 5M. There is no mention in the legend what are the meanings of P1, P2, P3, P4 and P5.
7. In figure 5J only a small fraction of the cells is stained for GFP (LRIM15) but in figure 5I the number of positive cells for LRIM15 is much higher. How that can be explained?
8. Figure 5h does not match figure 5i. In the UMAPs CFP positive cells concentrate in P1.2, P2.2 and some P3.3 clusters, however in figure 5i indicates that most of the cells that are CFP+ are concentrated in clusters P5.1, P4.3, P3.3 and P5.2. We believe that there was a mistake in figure 5i because figure 5j also corroborates figure 5h where cells that are PPO6 positive do not overlap with LRIM15. The proportion of positive cells in cluster P5.1 in Fig. 5h is much less than 40%.
9. This result indicates that PPO6 and LRIM15 are mostly expressed in different hemocyte subpopulations, with only a small proportion of cells expressing both reporters. They could possibly represent different types of hemocytes, for instance oenocytoids and granulocytes. The authors could do an immunofluorescence staining both markers (CFP and GFP) together with phalloidin staining to use cell morphology and spreading ability of granulocytes on a glass surface to define which cells are being detected by the PPO-CFP and LRIM15-GFP lines, respectively. This is critical to be able to use the transgenic lines in functional assays.
10. In figure 5J there is a typo on the legend. Blue color should be labelled as CFP+/GFP-, instead of CFP-/CFP-

Reviewer #3

(Remarks to the Author)

Reviewer #4

(Remarks to the Author)

The manuscript titled "Exploring new dimensions of immune cell biology in *Anopheles gambiae* through genetic immunophenotyping" by Samantsidis and Smith addresses knowledge gaps in the field of mosquito immunology. This is a commendable study that contributes significantly to mosquito immunology with implications for vertebrate immune system evolution. I congratulate the authors for their innovative use of imaging and genetic tools to advance our understanding of hemocyte populations. Addressing some minor revisions suggested below will further enhance the manuscript's impact and relevance.

Evaluation: The use of fluorescent protein reporters driven by hemocyte-specific promoters is an excellent strategy that partially compensates for the lack of highly specific markers for individual cell types in mosquitoes. This approach is well-conceived and executed. The application of spectral imaging flow cytometry reveals a remarkable diversity among hemocytes, identifying twelve distinct populations. This significantly extends current knowledge of mosquito hemocyte biology and highlights the complexity and plasticity of these immune cells.

This study sets a solid foundation for further investigations into mosquito immune responses, vector competence, and the evolutionary comparison of immune systems.

Suggested (minor) revisions: I recommend addressing the following minor concerns to strengthen the discussion and broaden its impact.

1. It would be beneficial to include a more comprehensive discussion of the study's limitations. For example, while the use of fluorescent reporters is commendable, the potential for incomplete specificity of promoters or overlap between markers in different cell populations could be explored further.

2. Adding a few sentences to contextualize how these findings might inform our understanding of immune system evolution, especially in comparison to vertebrate immune systems, would enhance the discussion. For example, the plasticity and diversity of hemocyte populations in mosquitoes could be linked to broader themes in innate immunity evolution.

Version 1:

Reviewer comments:

Reviewer #2

(Remarks to the Author)
Please see attachement.

Reviewer #3

(Remarks to the Author)
I co-reviewed this manuscript with one of the reviewers who provided the listed reports. This is part of the Nature Communications initiative to facilitate training in peer review and to provide appropriate recognition for Early Career Researchers who co-review manuscripts.

Reviewer #4

(Remarks to the Author)
The authors have successfully addressed my comments. The revised version of the manuscript has incorporated the necessary changes. I endorse the publication of this study.

In cases where reviewers are anonymous, credit should be given to 'Anonymous Referee' and the source. The images or other third party material in this Peer Review File are included in the article's Creative Commons license,

Response to Reviewer's comments

NCOMMS-24-70517

"Exploring new dimensions of immune cell biology in *Anopheles gambiae* through genetic immunophenotyping"

A detailed response to each of the reviewer comments is listed below. All changes in response to the reviewer's comments are highlighted in the text of a "highlighted changes" manuscript file uploaded as a Related Manuscript File.

Reviewer #2

This manuscript examines hemocyte populations considering morphology, ploidy, phagocytic capacity, and expression of hemocyte-specific genetic markers. The authors introduce the regulatory region of genes that were either expressed in all hemocytes (NimB2, PPO6 and SPARC) and some with granulocyte-specific expression (LRIM15 and SCRASP1) to drive expression of reporter genes (CFP or GFP) in specific hemocyte populations. The establishment of *Anopheles gambiae* transgenic mosquitoes that express reporter genes in specific hemocyte populations and the analysis of DNA content in hemocytes are important contributions. However, the transgenic lines had limited success, as two transgenic lines (NimB2-CFP and SCRASP1) had very low expression and could not be used. PPO6-CFP and SPARC-CFP are not "panspecific" hemocyte markers, as the fluorescence in adult females was mostly detected in granulocytes. Furthermore, although immunostainings with the LRIM15-GFP line detected phagocytic cells with low or high levels of GFP protein expression in adult females and the enrichment in mRNA levels between hemocytes and carcass was modest. The manuscript is well written, but it is complex and difficult to follow. A summary table indicating the expression level, hemocyte populations where the marker was expressed, the level of enrichment in hemocytes, the percentage of cells that express the marker, etc. would be very useful.

Major Points

1. The weakest point of the manuscript is the limited integration of the classification of hemocyte subpopulations defined by DNA ploidy and morphology with the information obtained from the fluorescent markers and the three major functional classes of hemocytes known to be present in adult females (prohemocytes, oenocytoids and granulocytes). The authors state: "Our results comprehensively map the composition of mosquito hemocytes, classifying them into twelve distinct populations based on size, granularity, ploidy, phagocytic capacity, and the expression of PPO6, SPARC, and LRIM15 genetic markers." However, the transgenic lines were not used to define the twelve populations and it is not clear how the reporter genes can help to distinguish the subpopulations. They define 12 subpopulations of hemocytes based on flow cytometry analysis of ploidy, size and granularity, but it is not clear what type of cell each of those subpopulations represent. Are they prohemocytes, oenocytoids or granulocytes? What is the functional importance of that classification?

Thank you for the comment. In our revised manuscript we have made significant attempts to be more transparent in the presentation of our flow cytometry analysis using the PPO6, SPARC, and LRIM15 lines. This includes modifications to the abstract and **Figure 5**, where we now display the individual cell populations labeled with CFP, GFP, or both markers when in the context of the LRIM15/SPARC or LRIM15/PPO6 backgrounds. In addition, we also summarize the morphological, phagocytic, and genetic properties of our cell types in what is now included in Table 1 to make our data more accessible and insightful regarding the cell populations identified in our analysis. With this summarized data, we provide classifications of each presumed hemocyte subtype to the best of our abilities, yet these are not absolute, and believe that the transparency in the presentation of our data in Table 1 also enable others in the community to come to their own assertions and conclusions.

Furthermore, the fluorescent proteins are only expressed in a low proportion of cell in the flow cytometry experiments (LRIM15, 15%; SPARC, 27 %; PPO6, 8.7%). In theory, if LRIM15 and SPARC are both granulocyte markers, so they should stain a similar proportion of cells. Furthermore, PPO6 was selected as a “panhemocyte” marker, but only stains a very small proportion of cells (8.7%). The low detection level could also result from sensitivity of the flow assays to detect fluorescence in smaller cells (prohemocytes and oenocytoids).

Thank you for the comment. As the reviewer points out, it would have been ideal if our markers were able to label a larger proportion of cells in our flow cytometry experiments. Our intended “panhemocyte” promoters, which were identified in part by previous scRNA-seq analysis, labeled a fraction of the total cell population, suggesting potential discordance between RNA and protein expression.

Moreover, we took a very conservative approach for our gating strategies employed in our flow cytometry experiments to get rid of any potential background signals that could be misconstrued as a false positive in our analysis. There is little doubt that this also excluded several “false negative” cells that had low levels of expression that could not readily be separated from the background signal. As a result, this approach likely decreased the percentage of positive cells identified through these flow cytometry experiments as compared to analysis using a hemocytometer and fluorescence microscopy. These results are now summarized in **Table S2**.

As suggested by another reviewer, we have added text to the discussion to address the limitations (such as these) of our study in our revised manuscript.

- 2. It will be informative to correlate the levels of fluorescent protein and the target protein itself. For example, the authors could do a double immunofluorescence staining of the PPO6-CFP line with an anti-CFP antibody together with an antibody to detect endogenous PPO6 expression. This could help to elucidate whether the promoter region that was chosen does not drive expression of the reporter that matches that of the endogenous gene. It is also possible that the mRNA expression used to select the markers does not match protein expression. For example, the mRNA could be present in all hemocytes, but only be transcribed in a specific subpopulation. Alternatively, the gene that was selected may have a general low level of expression and can only be detected in cells with higher ploidy.**

Thank you for the suggestion. We have performed these experiments as suggested using CFP and PPO6 antibodies to examine our PPO6-CFP line, with the results now displayed in **Figure S6** of our revised manuscript. Through these experiments, we demonstrate that ~75% of PPO6+ cells are also CFP+, suggesting that the cells with the PPO6-CFP construct accurately represent endogenous PPO6 expression. However, this is not absolute, and may be the result of inadequate regulatory regions in the PPO6 construct to drive high CFP expression in all cell types.

As the reviewer suggests, there could also be some differences between mRNA and protein expression in mosquito hemocytes, as was suggested previously (PMID: 27624304). However, questions of potential post-transcriptional regulation in mosquito required more focused study in the future. While this could also be influenced by ploidy, at least in the context of our current study, the bulk of PPO6+ cells (as well as SPARC+ and LRIM15+ cells) are of lower DNA content (ploidy). These data are now more transparent through data presentation in **Figure 5** and **Table 1** of our revised manuscript.

- 3. The five levels of DNA ploidy that the authors identified by flow cytometry is very interesting. However, the methods section for the flow cytometry is not very detailed, and it is hard to understand how some of the assays were performed. Specially it is not clear how flow cytometry was used to detect cell ploidy. What were the standards used to determine cell ploidy? The presence of cells of with different sizes (small, medium and large) for P3 and P4 and (small and large) for P1, P2 and P5 hemocytes suggests that DNA replication takes place in the smaller cells, presumably prohemocytes, and precedes hemocyte differentiation. Are P3.1 prohemocytes that give rise to P3.3 phagocytic granulocytes, with P3.2 being an intermediate stage of differentiation??**

Thank you for the comment. We have added additional details to the methodology in our revised manuscript regarding our flow cytometry experiments. We believe that these changes should improve transparency in the multiple measurements obtained through our flow cytometry data. This includes measurements of DNA content (for which we refer to as ploidy) that are now displayed in **Table 1** for each cell subpopulation that are based on the intensity of DRAQ5 that serve as an indirect measure of cell ploidy. Yet, it should be noted that we did not incorporate standards of diploid, tetraploid, or aneuploid cells in our flow cytometry experiments to directly assess chromosome number. For this reason, we avoid making additional claims as to copy numbers associated with these cells.

However, based on the intensity of DRAQ5 now provided in **Table 1** for our cell populations, we do see an approximate doubling in DRAQ5 signal between P1, P2, P3, and P4 cell populations that would infer differences in euploid and polyploid cells.

As the reviewer suggests, these data provide insight into potential mechanisms of mosquito immune replication and differentiation. We would argue that we identify likely prohemocyte, oenocytoid, granulocyte, and intermediate cell populations, yet the progression of these cell lineages remains elusive. It is of significant interest to further examine how these immune cell populations may change in response to blood-feeding or immune challenge, and hope to further delineate these lineages in the future.

- 4. LRIM15 and SPARC are only expressed in cells with higher ploidy and in larger cells (P5.1 P5.2, P4.3 and P3.3). Does this reflect low sensitivity of the fluorescence detection by flow cytometry, so that only cells with multiple copies of the genome and/or that are larger express enough protein to be detected?**

Actually, our data would suggest the opposite. The majority of PPO6+, SPARC+, and LRIM15+ cells are found in P4 and P5, where the lower DRAQ5 intensity (as compared to other clusters such as P1 and P2) would suggest that these are likely populations of euploid cells.

Minor Points

- 1. Using the single cell analysis to choose universal markers might not be the best strategy considering that expression is inflated and relative to the level of expression in all other cell in the experiment. For example, if a gene has relatively low counts in a cluster of cells and no expression on the rest of the cells, it will have a high fold change, but the positive cells express low levels of that gene, that is highly specific. It raises the question if the markers were chosen based on expression level or specificity.**

Our strategy for the selection of our promoter candidates included the incorporation of existing mosquito hemocyte literature, ranging from functional studies, to single-cell and proteomic studies. As a result, these promoter candidates were chosen both based on transcript expression from scRNA-seq studies, as well as their specificity to denote the majority of cells or specifically that of phagocytic granulocyte populations. Additional details have been added to our revised manuscript to reflect these inputs into our selection of candidate promoters.

Ultimately, these types of promoter expression experiments are “trial and error” and require future iteration. These includes attempts to refine the regulatory regions on our current promoters and the identification of potential alternative candidates to further improve upon our cell annotations.

- 2. It is very hard to see the fluorescence signal in larvae when it is shown in “blue” (Fig. 1a& 1b), please modify the image to make hemocytes more visible by either making them green or cyan.**

We have modified the images in Figure 1 to reflect those changes, eliminating the DIC channel and by altering the color, to hopefully make the expression patterns more clearly visualized in our revised manuscript.

- 3. Figures 1g-h and 2g-l measure the enrichment of CFP and GFP in the hemocytes of transgenic lines by qPCR but the interpretation might be misleading. The enrichment does not mean hat all hemocytes express the fluorescent marker but could reflect a small portion of hemocytes expressing high amounts of the mRNA of the fluorescent protein. This is further confirmed by microscopy after calculating the percentage of cells that express CFP.**

The intention of these experiments is to examine the specificity/enrichment of our transgenic constructs to hemocytes as compared to other mosquito tissues. We do not make any claims that our qPCR approach can estimate the proportion of fluorescent-positive hemocytes.

- 4. Considering the percentage of cells that express the fluorescent protein in all 3 mosquito lines, indicates that expression is happening in a subset of hemocytes instead of all.**

While it may have been our original intention to identify promoters that would serve as “universal-” and “granulocyte-specific” markers, it would appear that the PPO6, SPARC, and LRIM15 each predominantly label populations of granulocytes. These data are clearly presented in **Table 1** of our revised manuscript.

Although PPO6+, SPARC+, and LRIM15+ cells predominantly label granulocyte populations, there are however, differences in their abundance (**Figure 3** and **Table S2**). Additional co-expression analysis of CFP and GFP constructs expressing both fluorescent markers supports that LRIM15+/SPARC+ and LRIM15+/PPO6+ populations are unique and only represent a subset of fluorescently labeled cells (**Figure 5**), supporting that there is further complexity in the expression of these markers in mosquito granulocyte populations.

- 5. Low and high signals are hard to see in figures 2E. Please remove the DIC background placing an additional column in the panel.**

This figure has been modified in our revision to hopefully improve its visualization.

- 6. Some figure legends lack crucial information regarding the figures. For instance: Figure 5K and 5M. There is no mention in the legend what are the meanings of P1, P2, P3, P4 and P5.**

The figure legends in our revised manuscript have been substantially revised. This includes the referred to data now presented in **Figure 5g** and **5h**, where we have altered the presentation to provide specific data of each of the 12 hemocyte subtypes.

- 7. In figure 5J only a small fraction of the cells is stained for GFP (LRIM15) but in figure 5I the number of positive cells for LRIM15 is much higher. How that can be explained?**

In our revised manuscript we present data in **Figures 5g** and **5h** that are more representative of the CFP+/GFP-, CFP-/GFP+, and CFP+/GFP+ cell types. This also includes in data display that highlight specific immune cell subtypes representative of each of the above genetic phenotypes.

In addition, experiments that were performed with the LRIM15/PPO6 line were likely transheterozygous, meaning that not all mosquitoes would have had been homozygous for either GFP or CFP markers, potentially reducing the proportion of labeled cells.

- 8. Figure 5h does not match figure 5i. In the UMAPs CFP positive cells concentrate in P1.2, P2.2 and some P3.3 clusters, however in figure 5i indicates that most of the cells that are CFP+ are concentrated in clusters P5.1, P4.3, P3.3 and P5.2. We believe that there was a mistake in figure 5i because figure 5j also corroborates figure 5h where cells that are PPO6 positive do not overlap with LRIM15. The proportion of positive cells in cluster P5.1 in Fig. 5h is much less than 40%.**

Similar to the above response, we did revise the UMAPs and graphical data presented in **Figures 5g** and **5h** to better illustrate the specific subpopulations displaying CFP, GFP, or both markers. We believe that these data closely corroborate other data presented in our manuscript.

- 9. This result indicates that PPO6 and LRIM15 are mostly expressed in different hemocyte subpopulations, with only a small proportion of cells expressing both reporters. They could possibly represent different types of hemocytes, for instance oenocytoids and granulocytes. The authors could do an immunofluorescence staining both markers (CFP and GFP) together with phalloidin staining to use cell morphology and spreading ability of granulocytes on a glass surface to define which cells are being detected by the PPO-CFP and LRIM15-GFP lines, respectively. This is critical to be able to use the transgenic lines in functional assays.**

Thank you for the suggestion. We believe that changes made in our revised manuscript, such as the inclusion of **Table 1**, to better define hemocyte subtypes. This includes definitions of each individual cell type that display morphological properties, DNA content, phagocytic abilities, and the proportion of CFP/GFP cells with each individual promoter construct. In addition, we have also made modifications to **Figure 5** to better display the co-expression of CFP and GFP markers in immune cell subtypes. From these combined analyses, it is clear that PPO6+- and LRIM15+-positive cells exhibit properties of different granulocyte populations.

As a result, we believe that these quantitative measurements relying on flow cytometry far outweigh the gains of the suggested immunofluorescence experiments, and offer greater rigor and reproducibility.

- 10. In figure 5J there is a typo on the legend. Blue color should be labelled as CFP+/GFP-, instead of CFP-/CFP-.**

Thank you for catching this. This has been corrected in our revised manuscript.

Reviewer #3

Thank you for your efforts in the review of our manuscript. We appreciate your time and effort in the review of our manuscripts and hope that this was a positive and worthwhile experience.

Reviewer #4

The manuscript titled "Exploring new dimensions of immune cell biology in *Anopheles gambiae* through genetic immunophenotyping" by Samantsidis and Smith addresses knowledge gaps in the field of mosquito immunology. This is a commendable study that contributes significantly to mosquito immunology with implications for vertebrate immune system evolution. I congratulate the authors for their innovative use of imaging and genetic tools to advance our understanding of hemocyte populations. Addressing some minor revisions suggested below will further enhance the manuscript's impact and relevance.

Evaluation: The use of fluorescent protein reporters driven by hemocyte-specific promoters is an excellent strategy that partially compensates for the lack of highly specific markers for individual cell types in mosquitoes. This approach is well-conceived and executed. The application of spectral imaging flow cytometry reveals a remarkable diversity among hemocytes, identifying twelve distinct populations. This significantly extends current knowledge of mosquito hemocyte biology and highlights the complexity and plasticity of these immune cells.

This study sets a solid foundation for further investigations into mosquito immune responses, vector competence, and the evolutionary comparison of immune systems.

Suggested (minor) revisions: I recommend addressing the following minor concerns to strengthen the discussion and broaden its impact.

Minor Points

1. **It would be beneficial to include a more comprehensive discussion of the study's limitations. For example, while the use of fluorescent reporters is commendable, the potential for incomplete specificity of promoters or overlap between markers in different cell populations could be explored further.**

Thank you for the suggestion. We have amended the discussion section of our revised manuscript to incorporate a discussion of the current limitations of our study, which ultimately requires a further refinement to the novel flow cytometry techniques described in our manuscript and the need for additional immune cell markers to refine our characterization of mosquito hemocytes.

2. **Adding a few sentences to contextualize how these findings might inform our understanding of immune system evolution, especially in comparison to vertebrate immune systems, would enhance the discussion. For example, the plasticity and diversity of hemocyte populations in mosquitoes could be linked to broader themes in innate immunity evolution.**

Thank you again for the suggestion. We have added some brief text to the summary paragraph of our revised discussion to include the potential applications for comparative immunology through our described technological advances in the genetic resources and methodologies for the study of mosquito immune cells.

Response to Reviewer's comments

NCOMMS-24-70517A

"Characterization of *Anopheles gambiae* immune cells through genetic and functional immunophenotyping"

A detailed response to each of the reviewer comments is listed below. All changes in response to the reviewer's comments are highlighted in a manuscript file displaying "tracking changes" which was uploaded as a Related Manuscript File.

Reviewer #2

The authors addressed all inquiries in their rebuttal letter by providing new figures and experiments whenever applicable. They have resolved all the major concerns raised during the review process. However, there are some minor points that should be addressed to improve the clarity of their results.

One significant improvement was the addition of the modified Table 1, which facilitates interpretation of the results and enhances transparency. We believe the data would be more comprehensible if the authors reorganized the table to group the "presumed hemocyte subtypes" together and then ordered them by DNA content intensity measured by DRAQ5 (from high to low). Shading the DNA content proportionally to the DNA level, similar to what they did for the proportion of cell expressing the transgenes, would be very useful (see example below).

Table 1. Summary of gated hemocyte subpopulations.

Populati	Presumed subtype	% of cells*	FSC (M ± SE)	SSC (M ± SE)	DRAQ5 (M ± SE)	Phagocytic	%PPO6+	%SPARC+	%LRIM15+
P1.1	Oenocytoid	2.0	7,166 ± 650	2,936 ± 298	4,866,667 ± 121,975	-	0.1	0.0	0.1
P2.1	Oenocytoid	5.6	5,201 ± 949	2,255 ± 272	2,040,000 ± 60,000	-	0.1	0.1	0.3
P3.1	Prohemocyte	3.3	2,486 ± 28	1,011 ± 190	964,667 ± 35,951	-	0.0	0.2	0.0
P4.1	Prohemocyte	27.7	2,397 ± 71	781 ± 88	449,333 ± 15,344	-	0.5	0.0	0.0
P3.2	Intermediate	3.2	8,064 ± 528	2,810 ± 288	978,833 ± 27,668	+	2.3	1.1	1.3
P4.2	Intermediate	14.6	8,300 ± 432	2,852 ± 59	474,667 ± 18,800	+	4.2	7.1	4.9
P1.2	Granulocyte	0.9	31,844 ± 4,088	17,247 ± 2,744	4,316,667 ± 156,241	++	4.8	1.4	2.9
P2.2	Granulocyte	2.2	38,393 ± 528	12,392 ± 2,417	2,356,667 ± 57,831	++	7.1	2.7	4.1
P3.3	Granulocyte	3.0	24,387 ± 822	8,212 ± 139	1,040,000 ± 30,000	+++	12.7	6.2	9.9
P4.3	Granulocyte	3.0	26,046 ± 1,648	7,889 ± 494	433,506 ± 51,132	+++	17.6	13.7	12.5
P5.1	Granulocyte	30.5	13,982 ± 1,642	3,623 ± 237	85,220 ± 1,832	+/+	39.9	61.2	58.3
P5.2	Granulocyte	4.0	41,125 ± 2,717	14,606 ± 1,988	56,426 ± 1,915	+/+++	10.7	6.3	5.7

1. -*, percentage of total immune cells gated into hemocyte subpopulations
2. -Forward Scatter (FSC), Side Scatter (SSC), and nuclear staining (DRAQ5) intensity are presented as mean ± SE values from three biological replicates.
3. -Phagocytic capacity of hemocyte subtypes classified based on bead uptake rates as depicted in Fig. 6e.

Thank you for the well-articulated suggestion. We have modified **Table 1** in our revised manuscript to reflect these suggested changes regarding the presumed hemocyte subtypes and their order reflecting DNA content. While we appreciate the use of color to highlight these changes, it is Nature policy that tables must be black and white, such that we cannot accommodate all of these suggest changes.

These modifications in the table highlight that each hemocyte subpopulation has at least two different levels of DNA, suggesting that all hemocytes have the ability to duplicate their DNA, which could indicate either proliferation or endoreplication. This should be briefly discussed in the manuscript.

We have integrated additional text into the discussion of our revised manuscript to accommodate this suggestion.

The authors have clarified that ploidy measurement with DRAQ5 is an indirect method, as it reflects the relative abundance of DNA content between cells, because it is not known what level of DNA corresponds to a diploid genome in a given hemocyte type. The authors should also add a comment on this point in the main text.

We have added additional text to the discussion for transparency to highlight our inability to fully define ploidy levels (diploid, tetraploid, etc.) based on our measurements of DNA content via DRAQ5.

Reviewer #3

We would like to thank the reviewer and hope that this was a positive and insightful experience.

Reviewer #4

The authors have successfully addressed my comments. The revised version of the manuscript has incorporated the necessary changes. I endorse the publication of this study.

We appreciate your positive endorsement of our work and thank you for your help in improving our study.

REVIEWER 2, ATTACHMENT 1

The authors addressed all inquiries in their rebuttal letter by providing new figures and experiments whenever applicable. They have resolved all the major concerns raised during the review process. However, there are some minor points that should be addressed to improve the clarity of their results.

One significant improvement was the addition of the modified **Table 1**, which facilitates interpretation of the results and enhances transparency. We believe the data would be more comprehensible if the authors reorganized the table to group the "presumed hemocyte subtypes" together and then ordered them by DNA content intensity measured by DRAQ5 (from high to low). Shading the DNA content proportionally to the DNA level, similar to what they did for the proportion of cell expressing the transgenes, would be very useful (see example below).

Table 1. Summary of gated hemocyte subpopulations.

Populati	Presumed subtype	% of cells*	FSC (M ± SE)	SSC (M ± SE)	DRAQ5 (M ± SE)	Phagocytic	%PPO6+	%SPARC+	%LRIM15+
P1.1	Oenocytoid	2.0	7,166 ± 650	2,936 ± 298	4,866,667 ± 121,975	-	0.1	0.0	0.1
P2.1	Oenocytoid	5.6	5,201 ± 949	2,255 ± 272	2,040,000 ± 60,000	-	0.1	0.1	0.3
P3.1	Prohemocyte	3.3	2,486 ± 28	1,011 ± 190	964,667 ± 35,951	-	0.0	0.2	0.0
P4.1	Prohemocyte	27.7	2,397 ± 71	781 ± 88	449,333 ± 15,344	-	0.5	0.0	0.0
P3.2	Intermediate	3.2	8,064 ± 528	2,810 ± 288	978,833 ± 27,668	+	2.3	1.1	1.3
P4.2	Intermediate	14.6	8,300 ± 432	2,852 ± 59	474,667 ± 18,800	+	4.2	7.1	4.9
P1.2	Granulocyte	0.9	31,844 ± 4,088	17,247 ± 2,744	4,316,667 ± 156,241	++	4.8	1.4	2.9
P2.2	Granulocyte	2.2	38,393 ± 528	12,392 ± 2,417	2,356,667 ± 57,831	++	7.1	2.7	4.1
P3.3	Granulocyte	3.0	24,387 ± 822	8,212 ± 139	1,040,000 ± 30,000	+++	12.7	6.2	9.9
P4.3	Granulocyte	3.0	26,046 ± 1,648	7,889 ± 494	433,506 ± 51,132	+++	17.6	13.7	12.5
P5.1	Granulocyte	30.5	13,982 ± 1,642	3,623 ± 237	85,220 ± 1,832	+/+	39.9	61.2	58.3
P5.2	Granulocyte	4.0	41,125 ± 2,717	14,606 ± 1,988	56,426 ± 1,915	++/+++	10.7	6.3	5.7

1. -, percentage of total immune cells gated into hemocyte subpopulations
2. -Forward Scatter (FSC), Side Scatter (SSC), and nuclear staining (DRAQ5) intensity are presented as mean ± SE values from three biological replicates.
3. -Phagocytic capacity of hemocyte subtypes classified based on bead uptake rates as depicted in Fig. 6e.

These modifications in the table highlight that each hemocyte subpopulation has at least two different levels of DNA, suggesting that all hemocytes have the ability to duplicate their DNA, which could indicate either proliferation or endoreplication. This should be briefly discussed in the manuscript.

The authors have clarified that ploidy measurement with DRAQ5 is an indirect method, as it reflects the relative abundance of DNA content between cells, because it is not known what level of DNA corresponds to a diploid genome in a given hemocyte type. The authors should also add a comment on this point in the main text.